# Geometry design of tethered small-molecule acceptor enables highly stable and efficient polymer solar cells

Yang Bai[1,12], Ze Zhang[1,12], Qiuju Zhou[2], Hua Geng[3], Qi Chen[1], Seoyoung Kim[4], Rui Zhang [5], Cen Zhang[1], Bowen Chang[1], Shangyu Li[1], Hongyuan Fu[1], Lingwei Xue[1], Haiqiao Wang[6], Wenbin Li[7], Weihua Chen [7], Mengyuan Gao[8], Long Ye[8], Yuanyuan Zhou [9], Yanni Ouyang[9], Chunfeng Zhang [10], Feng Gao [5], Changduk Yang [4], Yongfang Li [11] & Zhi-Guo Zhang [1]✉

With the power conversion efficiency of binary polymer solar cells dramatically improved, the thermal stability of the small-molecule acceptors raised the main concerns on the device operating stability. Here, to address this issue, thiophene-dicarboxylate spacer tethered small-molecule acceptors are designed, and their molecular geometries are further regulated via the thiophene-core isomerism engineering, affording dimeric TDY-α with a 2, 5-substitution and TDY-β with 3, 4-substitution on the core. It shows that TDY-α processes a higher glass transition temperature, better crystallinity relative to its individual small-molecule acceptor segment and isomeric counterpart of TDY-β, and a more stable morphology with the polymer donor. As a result, the TDY-α based device delivers a higher device efficiency of 18.1%, and most important, achieves an extrapolated lifetime of about 35000 hours that retaining 80% of their initial efficiency. Our result suggests that with proper geometry design, the tethered small-molecule acceptors can achieve both high device efficiency and operating stability.

During the last 5 years, polymer solar cells (PSCs) have witnessed a significant progress with extensive investigations on donor-acceptor (D-A)-type small-molecule acceptors (SMA)[1–5], enabling a significant leap forward to achieve over 18% power conversion efficiency (PCE)[6–12].

For high-performance PSCs, active layers with a nanoscale bicontinuous-interpenetrating network of polymer donors and SMAs are required to facilitate exciton dissociation and charge carrier transport[13,14]. However, such deliberately tuned morphologies are

[1]State Key Laboratory of Chemical Resource Engineering, Beijing Advanced Innovation Center for Soft Matter Science and Engineering, Beijing University of Chemical Technology, Beijing 100029, China. [2]Analysis & Testing Center, Xinyang Normal University, Xinyang, Henan 464000, China. [3]Beijing Key Laboratory for Optical Materials and Photonic Devices, Department of Chemistry, Capital Normal University, Beijing 100048, China. [4]Department of Energy Engineering, School of Energy and Chemical Engineering, Low Dimensional Carbon Materials Center, Ulsan National Institute of Science and Technology (UNIST), Ulsan 689-798, South Korea. [5]Department of Physics, Biomolecular and Organic Electronics, Chemistry and Biology (IFM), Linköping University, Linköping SE-58183, Sweden. [6]Beijing Engineering Research Center for the Synthesis and Applications of Waterborne Polymers, Beijing University of Chemical Technology, Beijing 100029, China. [7]College of Chemistry & Green Catalysis Center, Zhengzhou University, Zhengzhou 450001, China. [8]School of Materials Science and Engineering, Tianjin Key Laboratory of Composite and Functional Materials, Tianjin University, Tianjin 300350, China. [9]Department of Physics, Hong Kong Baptist University, Hong Kong, China, Smart Society Lab, Hong Kong Baptist University, Hong Kong, China. [10]National Laboratory of Solid State Microstructures, School of Physics, and Collaborative Innovation Center for Advanced Microstructures, Nanjing University, Nanjing 210093, China. [11]Beijing National Laboratory for Molecular Sciences, CAS Key Laboratory of Organic Solids, Institute of Chemistry, Chinese Academy of Sciences, Beijing 100190, China. [12]These authors contributed equally: Yang Bai, Ze Zhang. ✉e-mail: zgzhangwhu@iccas.ac.cn

usually in a state of thermodynamic instability, thus they are with an obvious tendency of burn-in degradation during long-term operations or at elevated temperatures. The degradation is due to the inevitable thermodynamic relaxation of the mixed domains in the blend from their initially trapped state to the binodal states[15–20]. On the other hand, the classical record-holding A-DA'D-A type SMAs have the disadvantages of excessive crystallinity and relatively high miscibility with polymer donors that could lead to diffusion-induced deterioration of the morphology[21]. Thus, how to balance efficiency and stability is still the current main focus of molecular engineering on SMAs for reaching commercialization.

Recently, Ade et al. discovered that the diffusion of SMAs into the donor polymers exhibits Arrhenius behavior and that the diffusion coefficients decrease exponentially with increasing glass transition temperature ($T_g$) of the SMAs[21]. Their observation can well explain the high stability of all-PSCs that use polymerized SMAs (PSMAs) as acceptor[22,23] which is composed of SMAs as a skeleton structure linked with a variety of π-bridge linking units in the main chain[24–29]. The success of PSMAs also triggered an investigation on the conjugated dimeric SMAs to remove the disadvantage of batch-to-batch reproducibility of the PSMAs[30–37]. Prominent device efficiency with lower energy loss and high device stability was realized based on those dimeric SMAs, highlighting the importance of the dimeric approach[38]. Instead of using aromatic linker, alternatively, we covalently tethered SMAs (TSMAs) with flexible linkages on a benzene core[39]. Such an approach provides the acceptor with excellent solubility for device processing and renders their synthesis less depending on the toxicity of the organotin reagent as well as the noble metal catalyst. With flexible spacers to restrict the motion of individual SMAs, the TSMAs show a larger $T_g$ to suppress the thermodynamic relaxation in mixed domains, affording substantially reduced burn-in loss from their initial PCE of 17.85%. Those encouraging results suggest a plausible approach to construct PSCs simultaneously with high efficiency and excellent operating stability. Of critical importance is how to further regulate the photophysical properties of the TSMAs toward further improved device performance.

Recently, it has been well established that the electronic state coupling of the SMAs significantly affects the exciton splitting in devices[40–42]; thus, in the dimers, if the molecular architectures can be tuned to manipulate their molecular packing, more desirable photophysical properties can be expected. Considering that the aromatic central core can provide different locations of a spacer for attaching individual SMAs, the variation on the aromatic core would directly result in different molecular architectures and thus provide the possibility of tuning their aggregation, so long as the device performance. Our hypothesis is supported by a class of well-studied liquid crystal oligomers for which mesogenic units are also connected on the central aromatic core via flexible spacers, whereby the molecular topology determines their aggregation behavior[43,44]. In this scenario, it will be interesting to investigate the effect of molecular topology on their aggregation and relaxation behavior, thus the possibility of their device performance. With these considerations, here, two TSMA isomers based on two different thiophene dicarboxylate-based spacers are designed and synthesized. In our design, the central five-membered heterocyclic core is expected to promote the molecule for a bent shape rather than a linear shape as previously reported[39]. Furthermore, the molecular topology can be further tuned as the bend angle of the 2,5-thiophene dicarboxylate unit for TDY-α is typically 155.44° and the 3,4-thiophene dicarboxylate unit for TDY-β is typically 73.64° (Fig. 1a). The isomer effect provides the possibility to fully study the structure-properties with the variation on the molecular topology, affording TSMAs with different $T_g$ values, crystallization behavior and Flory–Huggins interaction parameter with the polymer donor. As a result, the TDY-α-based PSCs exhibit a further increased

efficiency as high as 18.1%, and more important, substantially reduced burn-in loss relative to that of Y6 and TDY-β.

## Results

### Materials synthesis and characterization

Chemical structures of the TSMAs are illustrated in Fig. 1b. And their synthetic routes are depicted in Supplementary Fig. 1, following a straightforward procedure in our recent publication toward TDY-α and TDY-β with high yields. Notably, for the last step, the dimers can be nearly quantitatively obtained via our newly developed $BF_3 \cdot OEt_2$-catalyzed Knoevenagel condensation[45], while using the classical pyridine-catalyzed Knoevenagel condensation only resulted in a low reaction yields of ca. 40%. For our design, the isomerism engineering on the aromatic core is expected to afford conformationally sensitive TSMAs, which could help to explore the relationship between molecular geometry and molecular aggregation behavior.

The 2D $^1H$-$^1H$ nuclear overhauser effect spectroscopy (NOESY) NMR spectra demonstrate an observed NOE correlation signal only between protons that are up to 5 Å apart, and is conducted to the investigation of the interaction of different molecules, especially the spatial relationship[46–48]. Here, such technology was used to probe the aggregation behavior of the Y6 subunits in the dimers (Supplementary Figs. 2–11). As expected, the 2D $^1H$-$^1H$ NOESY NMR spectra of monomeric Y6 under 298 K show several intrinsic intramolecular interactions with the off-diagonal signals (Supplementary Fig. 2), and there are no other correlation signals corresponding to the intermolecular interactions. However, for the dimers, the 2D $^1H$-$^1H$ NOESY NMR spectra reveal obvious additional off-diagonal signals (Fig. 1d, e, green-dashed box), which belong to the aromatic protons on the acceptor end-group (marked as $H^1$) and the alkyl chain on the pyrrole ring (marked as $H^2$). As no similar correlation signal for monomeric Y6 was observed (Fig. 1f, red-dashed box), the off-diagonal signals of TSMAs must be defined as the intramolecular interactions of the individual Y6 subunits in the dimers. This suggested that the Y6 subunits are folded. In this way, related protons can keep within 5 Å apart. Based on the NOESY result, the proposed geometry of TSMAs is given in Fig. 1c. Moreover, when the temperature was raised to 328 K, the off-diagonal signals for TDY-α disappeared (Supplementary Fig. 5), while that of TDY-β can still be observed (Supplementary Fig. 8). The results confirm a stronger intramolecular interaction of the Y6 subunits in TDY-β, and suggest that TDY-β is more stable under a more bent molecular shape.

The determination of the lowest energy structure of a chemical system is a very difficult problem due to the large number of local minima existing in its potential energy surface. In the search for the global minimum structure for different conformations, we make a preliminary geometric optimization through Forcite module using COMPASS force-field[49], and then semiempirical DFTB method[50] including dispersion correction to calculate the total energy as a function of the tilt angle of alkyl chain for the dimers (Fig. 1g, j). In our DFTB calculation, for the dimer with a linear fashion, its energy was set to zero. Based on this, the relative energy was provided for different geometries (Supplementary Figs. 12 and 13). Generally, the electrostatic attractions between electron-rich D-units and electron-deficient A-units result in the overlapping of individual Y6 subunits, thus delivering a folded geometry to refine the total energy in a dimer. Interestingly, there are essentially the lowest total energy geometric structures for both dimers: revealing 32.7° tilt angle for TDY-α system and 41.5° tilt angle for TDY-β system. Notably, since the transition from the initial state to the lowest energy point for TDY-α and TDY-β involves multiple conformations, we cannot simply conclude that a lower energy state for TDY-β necessarily implies that it is easier to fold than TDY-α. With a refined geometry, the top view and side view of the chemical structures of TDY-α are respectively provided in Fig. 1h, i, while those of TDY-β are respectively provided in Fig. 1k, l. It revealed

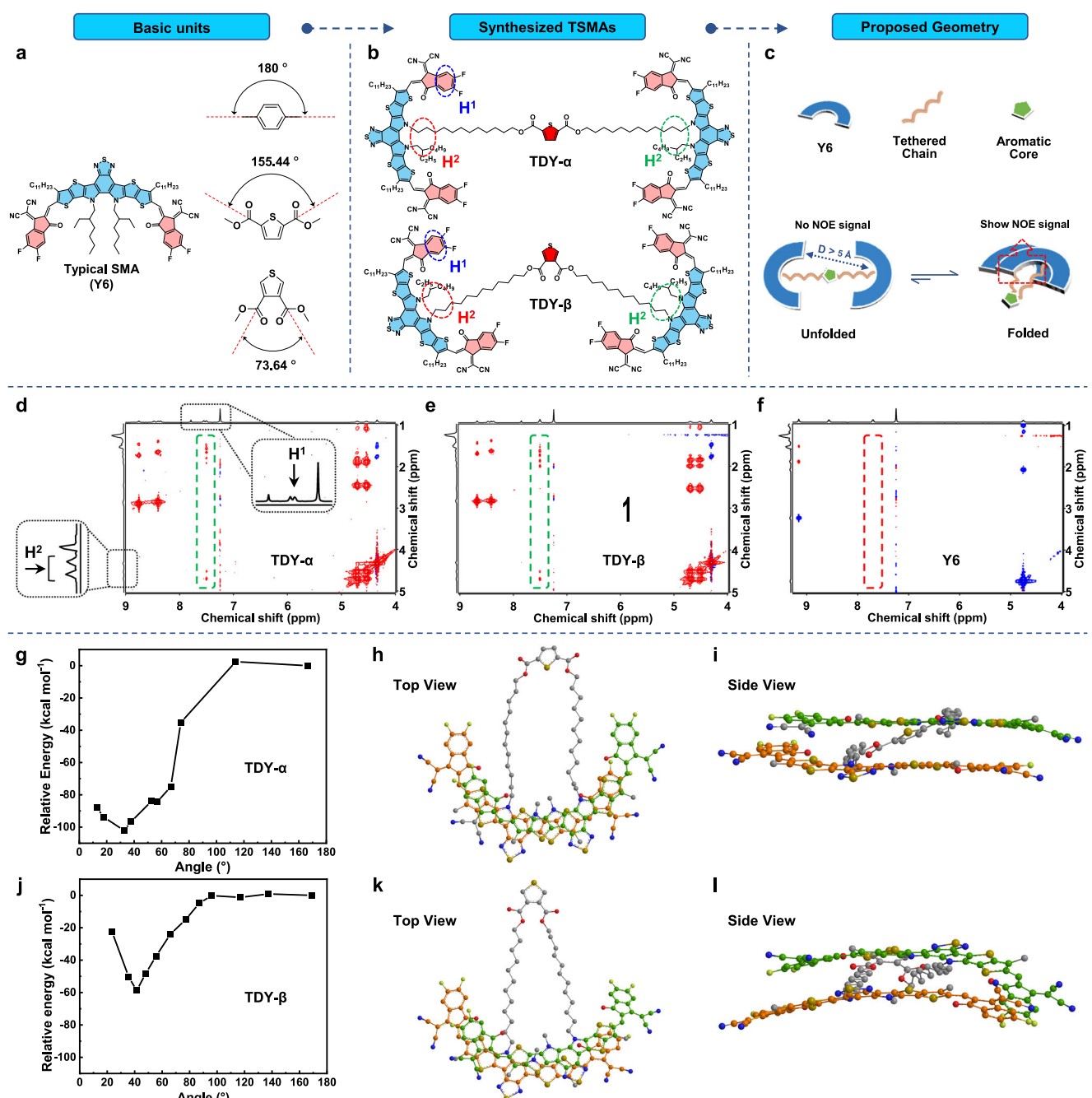

**Fig. 1 | Molecular geometry of the TSMAs. a** Basic units of TSMAs and the illustration of bend angles of 1,4-substituted benzene, 2,5-substituted thiophene and 3,4-substituted thiophene. **b** The molecular structures of the TSMAs. **c** Proposed geometry of TSMAs along with the illustration on the intramolecular NOE signals. 2D $^1$H-$^1$H NMR spectra of **d** TDY-α, **e** TDY-β and **f** Y6. The calculation of the total energy as a function of the tilt angle of alkyl chain via semiempirical DFTB method for **g** TDY-α and **j** TDY-β systems. The top view and side view of the optimal geometric configurations of **h**, **i** TDY-α and **k**, **l** TDY-β, respectively. The hydrogen atom, the outer and inner chains of the Y6 core are omitted for clarity.

that the H$^1$-H$^2$ distance of their geometries is smaller than 5 Å, consistent with our observation from the 2D $^1$H-$^1$H NOESY NMR spectra.

To probe the aggregation behavior of the TSMAs, we measured their UV-visible absorption spectra in solution at room temperature. In dilute solution, the UV-visible absorption spectra (Fig. 2a) of the TSMAs and Y6 show similar main absorption peaks centered at around 730 nm, while the TSMAs feature much stronger shoulder absorption peaks at around 693 nm. The obvious difference can also be observed even in ultra-dilute solution (Fig. 2c and Supplementary Fig. 14), suggesting the existence of some intrinsic aggregation in dilute solution for the dimers. Together with the conformations suggested by the

NOESY NMR spectra, our results suggest that the shoulder absorption peaks most likely originated from the electronic state coupling of individual SMAs in one dimer molecule. When the two Y6 subunits folded, their electronic states became coupled, leading to the Davydov splitting[51] of the original energy level for a new absorption band at 690 nm. As the strength of the coupling between the electronic states depends on the distance and orientation of the two Y6 subunits, the intensity of the 0–1 absorption peaks is slightly different between the dimers as shown in Fig. 2a[51,52]. Clear differences in the absorption are evident with increased temperature, as seen by the weakened Davydov splitting peaks observed for TDY-α in Supplementary Fig. 15. This

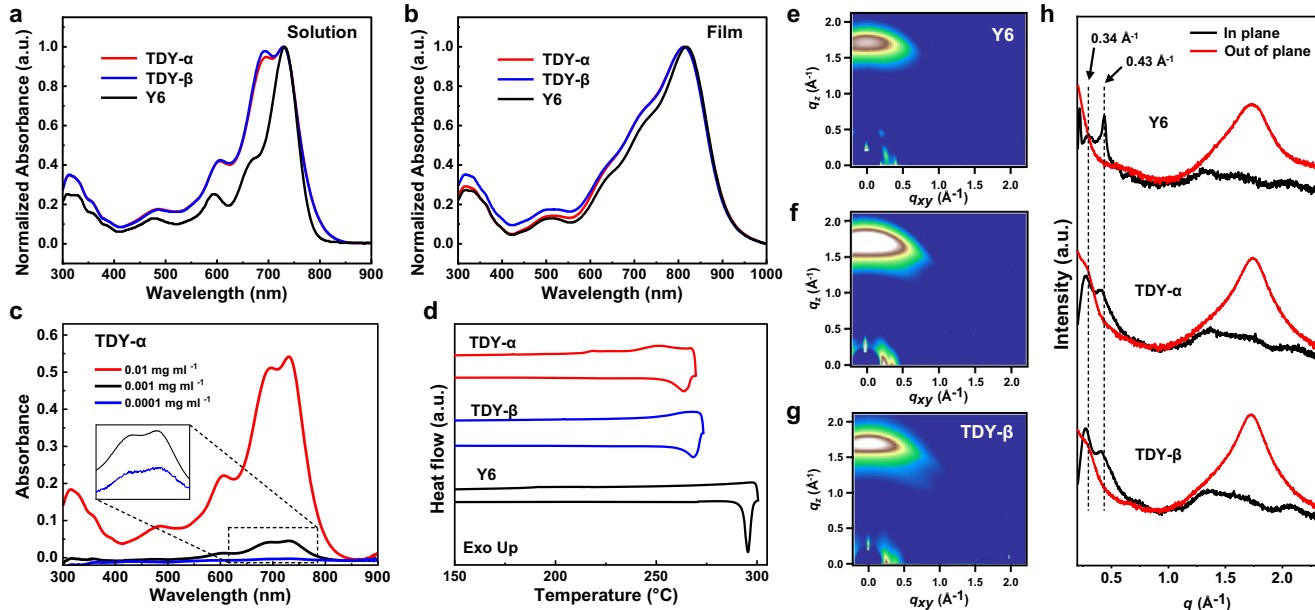

**Fig. 2 | Aggregation behavior of the dimers in solution and film.** Normalized UV-vis absorption spectra of the TSMAs and Y6 **a** in solution and **b** film. **c** The UV-vis absorption profiles of TDY-α in chloroform with various concentrations. **d** DSC thermograms of the TSMAs and Y6 with a heating rate of 10 °C min⁻¹ under a nitrogen atmosphere. The 2D GIWAXS diffraction patterns of the as-cast films of **e** Y6, **f** TDY-α and **g** TDY-β. **h** Line cuts of the GIWAXS images.

suggests a partial disaggregation of the dimers at a higher temperature. On the other hand, TDY-β appears to be more stable under a more bent molecular shape. These results are consistent with the NOESY results obtained at different temperatures.

It is well accepted that the pre-aggregation of acceptors in the solution can manipulate the morphology of the films[53–55]. For their film state (Fig. 2b), the main absorption peaks of TSMAs show a slightly hypochromatic shift of 6 nm relative to that of monomeric Y6, while the relative intensity of the shoulders of the Davydov splitting peaks (around 727 nm) also indicates a different extent of extra aggregation for the TSMAs. The result indicates a similar aggregation behavior between solution and films, and the pre-aggregation of the dimers in solution provides a plausible approach to tune the molecular packing behavior in films. The film absorption coefficient was measured, showing high values of $1.15 \times 10^5$ cm⁻¹ for TDY-α or $1.12 \times 10^5$ cm⁻¹ for TDY-β.

To further explore the crystallization behavior of TSMAs in films, Grazing-Incidence Wide Angle X-ray Scattering (GIWAXS) technique is used. It can be seen from Fig. 2e–g that both the dimers and Y6 in neat films favor a face-on orientation relative to the substrate, as revealed by the strong (010) π-π stacking peaks in the out-of-plane (OOP) direction and the (100) lamellar stacking peaks in the in-plane (IP) direction. For the dimers, despite the restriction on their molecular packing with a spacer, they show a similar facial π-π stacking distance (3.59 Å for TDY-α, 3.62 Å for TDY-β) relative to that (3.60 Å) of Y6. Interestingly, the dimers favor a larger crystal coherence length (25.7 Å for TDY-α, 28.3 Å for TDY-β) relative to that (23.6 Å) of Y6 (Supplementary Fig. 16). Furthermore, we calculated the relative degree of crystallinity (rDOC) of the three acceptors according to the GIWAXS data to quantitatively compare the crystallinity degree[21]. The obtained rDOC values of Y6, TDY-α, TDY-β are 0.19, 0.65, 1.00, respectively (Supplementary Fig. 17), which indicates a relatively higher crystallinity degree in tethered dimers. The higher degree of crystallinity may contribute in part to the high electron mobility ($\mu_e$) of the dimers ($3.65 \times 10^{-4}$ cm² V⁻¹ s⁻¹ for TDY-α, $3.12 \times 10^{-4}$ for TDY-β) relative to that of Y6 ($2.84 \times 10^{-4}$ cm² V⁻¹ s⁻¹).

The molecular packing of Y6 molecule in film is highly complex, comprising not only an overlap between A-end groups but also an

overlap between the central aromatic cores. The column stacking of the A-end groups creates an electron transport channel, with the distance between the A-end groups being referenced to the lamellar distance[21]. This periodic stacking structure can be accurately revealed by GIWAXS. In the context of Y6 in the IP direction, it has been observed that the two stacking peaks (0.34 and 0.43 Å⁻¹) are closely related to the lamellar distance between the A-end group stacking columns[41,56]. However, for the dimers, the lamellar distance tends to increase slightly with the presence of a linker (as depicted in Fig. 2h). As suggested in Fig. 1, our DFTB calculation reveals a distinct orientation of the Y6 subunits under a folded geometry. This can well explain the slightly varied stacking behavior and lamellar distance for the dimers in films. Therefore, studying the folding behavior of individual dimers can significantly aid in understanding the overall behavior of the bulky material.

For acceptors, when they are blended with polymer donors, their diffusion coefficients decrease exponentially with $T_g$. Here, the $T_g$ values are examined with the UV-vis deviation metric results following the method provided by Ade et al.[19]. As expected, with the spacer to restrict the motivation of the Y6 subunits, the dimers show a higher $T_g$ value (115 °C for TDY-α, 106 °C for TDY-β) relative to that of Y6 (99 °C) (Supplementary Fig. 18). The higher $T_g$ value of TDY-α suggests a suppressed thermal relaxation of the molecular structure under a less bent molecular shape. From the differential scanning calorimetry (DSC) measurements (Fig. 2d), they exhibit decreased melt points of 263.7 °C for TDY-α, 268.4 °C for TDY-β, with melting enthalpy ($\Delta H_m$) of 13.40 and 8.72 J g⁻¹, respectively. Frontier orbital energy levels of the TSMAs and Y6 are estimated from electrochemical cyclic voltammetry (CV) measurements (Supplementary Fig. 19), with the results provided in Table 1 and Fig. 3b. Both TDY-α and TDY-β exhibit a higher lowest unoccupied molecular orbital (LUMO) energy level ($E_{LUMO}$) values compared to Y6, which favors a larger open circuit voltage ($V_{OC}$) in devices. Furthermore, from Table 1, we can see that the dimers exhibit higher decomposition temperatures compared to Y6. Among the dimers, TDY-α shows a slightly higher decomposition temperature than TDY-β, which may be attributed to the higher bond energy of the carboxylate group attached to the thiophene core (319.81 kJ mol⁻¹ for TDY-α vs 313.65 kJ mol⁻¹ for TDY-β) (Supplementary Fig. 20).

**Table 1 | The physicochemical and aggregated properties of TSMAs and Y6**

| Acceptors | $\lambda_{max}$ (nm)[a] | $\lambda_{max}$ (nm)[b] | $E_g^{opt}$ (eV)[c] | $E_{LUMO}$ (eV)[d] | $E_{HOMO}$ (eV)[d] | $d$ [010] (Å) | CCL[010] (Å)[e] | rDOC[f] | $T_d$ (°C)[g] | $T_m$ (°C) | $\Delta H_m$ (J g⁻¹) | $T_g$ (°C)[h] | $\chi$ |
|---|---|---|---|---|---|---|---|---|---|---|---|---|---|
| Y6 | 731 | 819 | 1.326 | −4.09 | −5.70 | 3.60 | 23.6 | 0.19 | 318 | 295.5 | 28.15 | 99 | 0.89 |
| TDY-α | 730 | 813 | 1.332 | −3.92 | −5.69 | 3.59 | 25.7 | 0.65 | 331 | 263.7 | 13.40 | 115 | 0.98 |
| TDY-β | 729 | 812 | 1.337 | −3.96 | −5.76 | 3.62 | 28.3 | 1.00 | 328 | 268.4 | 8.72 | 106 | 1.09 |

[a]Solution.
[b]Film.
[c]Calculated from the absorption edge of the films: $E_g^{opt} = 1240/\lambda_{edge}$.
[d]Calculated according to the equation: $E_{LUMO/HOMO} = -e\,(\varphi_{red/ox} + 4.366)$ (eV).
[e]Calculated from Scherer equation: $CCL = 2\pi K/\Delta q$, where $\Delta q$ is the full-width at half-maximum of the peak and $K$ is a shape factor (0.9 was used here).
[f]Calculated the relative degree of crystallinity (rDOC) by integrating the scattering intensity of the lamellar diffraction peaks.
[g]The onset thermal decomposition temperatures with 5% weight loss.
[h]Estimated with absorption spectroscopy.

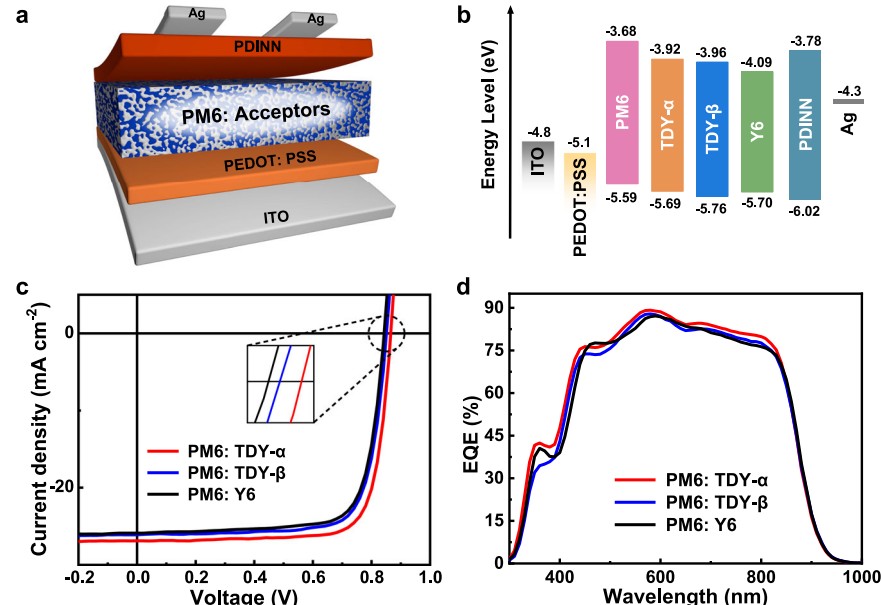

**Fig. 3 | Device performance of the TSMAs. a** Device structure of the PSCs. **b** Energy level diagram of the device. **c** *J*–V curves of the best PSCs with D:A weight ratio of 1:1.2 under the illumination of AM 1.5G, 100 mW cm⁻² and **d** EQE spectra of corresponding PSCs.

## Photovoltaic properties

PSCs are fabricated with a conventional architecture (Fig. 3a), where PM6 was used as the donor and PDINN (aliphatic amine-functionalized perylene-diimide)[57] was used as cathode interlayer facilitating carrier collection. The current–density–voltage (*J*–V) curves are shown in Fig. 3c, and photovoltaic parameters of the devices are collected in Table 2. The control devices based on monomeric Y6 possess a typical PCE of 16.2% with a $V_{OC}$ of 0.84 V, a $J_{SC}$ of 25.9 mA cm⁻², and an FF of 74.3%. In contrast, the PM6: TDY-α based devices show the best photovoltaic performance, with the highest PCE of 18.1% along with a $V_{OC}$ of 0.864 V, a $J_{SC}$ of 26.9 mA cm⁻², and an FF of 78.0%. While for its isomer counterpart of TDY-β, only a moderate PCE of 17.0% was obtained. The accuracy of the observed $J_{SC}$ values was confirmed by integrating the external quantum efficiencies (EQE) spectra with the AM 1.5G solar spectrum, showing a discrepancy of less than 5% (Fig. 3d). Compared with the Y6-based devices, higher $V_{OC}$ values are achieved for the TSMAs, ascribing to the upshift of $E_{LUMO}$ and an increase in the bandgap energy as suggested by our previous results.

To study the charge transport properties of the TSMAs, the carrier mobilities are calculated by measuring the single-carrier devices with the space charge limited current (SCLC) method. As presented in Supplementary Fig. 21 and Table 2, their respective hole/electron mobilities ($\mu_h/\mu_e$) of the PM6: TDY-α, PM6: TDY-β and PM6: Y6 blend films are $5.88 \times 10^{-4}$ cm² V⁻¹ s⁻¹/$4.92 \times 10^{-4}$ cm² V⁻¹ s⁻¹, $5.86 \times 10^{-4}$ cm² V⁻¹ s⁻¹/$4.69 \times 10^{-4}$ cm² V⁻¹ s⁻¹ and $5.32 \times 10^{-4}$ cm² V⁻¹ s⁻¹/$4.20 \times 10^{-4}$ cm² V⁻¹ s⁻¹, which correspond to $\mu_h/\mu_e$ ratios of 1.19, 1.25, and 1.27, respectively. The high and more balanced charge transport in the TDY-α based blend film was conducive to reducing charge accumulation and recombination, thus endowing its device with a higher FF.

To gain insight into exciton dissociation and charge collection behavior of the PSCs, the charge dissociation probability (P(E,T)) is estimated with the relationship between photocurrent density ($J_{ph}$) and the effective voltage ($V_{eff}$) of the TSMA-based devices (Supplementary Fig. 22). It demonstrated an improved exciton dissociation and charge collection process for the TSMAs, which is in line with a higher $J_{SC}$ value in the TSMA-based devices. Furthermore, the charge recombination behavior of the PSCs is investigated by the dependence of $V_{OC}$ and $J_{SC}$ on light intensity ($P_{light}$) with the results provided in Supplementary Fig. 23. It demonstrated that there is effective carrier collection and negligible bimolecular recombination at the short-circuit condition, which can also explain its larger $J_{SC}$ and FF values of the TDY-α based devices.

## Aggregation behavior of the blend films

The influence of the isomer effect on molecular packing and crystal texture in the blend films was investigated by GIWAXS measurement. As shown in Fig. 4a–c, all three blend films adopt a preferred face-on orientation, as evidenced by the strong lamellar stacking (100)

**Table 2 | Photovoltaic parameters of the PSCs based on PM6:TSMAs with D:A weight ratio of 1:1.2 and thermal annealing at 100 °C for 5 min under the illumination of AM 1.5G, 100 mW/cm² ᵃ**

| Active layer | $V_{oc}$ (V) | $J_{sc}$ (mA cm$^{-2}$) | FF (%) | PCE (%) | $\mu_h$ (10$^{-4}$cm$^2$V$^{-1}$s$^{-1}$) | $\mu_e$ (10$^{-4}$cm$^2$V$^{-1}$s$^{-1}$) | $\mu_h/\mu_e$ |
|---|---|---|---|---|---|---|---|
| PM6:TDY-α | 0.864 (0.858 ± 0.003) | 26.9 (26.4 ± 0.4) | 78.0 (77.7 ± 0.2) | 18.1 (17.7 ± 0.2) | 5.88 | 4.92 | 1.19 |
| PM6:TDY-β | 0.849 (0.851 ± 0.003) | 26.1 (25.7 ± 0.5) | 76.6 (75.1 ± 0.7) | 17.0 (16.4 ± 0.4) | 5.86 | 4.69 | 1.25 |
| PM6:Y6 | 0.842 (0.845 ± 0.004) | 25.9 (25.3 ± 0.5) | 74.3 (73.5 ± 0.8) | 16.2 (15.7 ± 0.4) | 5.32 | 4.20 | 1.27 |

ᵃAverage values based on ten devices.

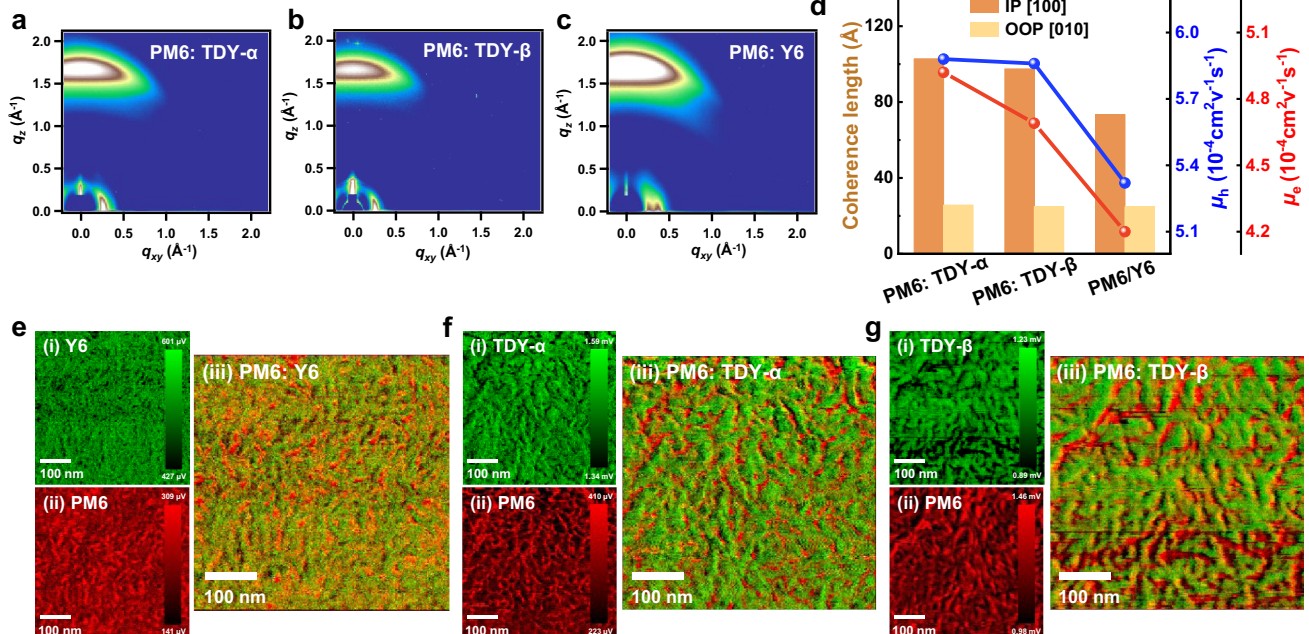

**Fig. 4 | Morphology and mobilities of the blends.** The 2D GIWAXS diffraction patterns of the blend films of **a** PM6:TDY-α, **b** PM6:TDY-β, and **c** PM6:Y6. **d** The corresponding crystal coherence lengths and the hole/electron mobilities of the blend films. PiFM images of the blend films. **e** Patterns of PM6:Y6 blend. **f** Patterns of PM6:TDY-α blend. **g** Patterns of PM6:TDY-β blend. (i) PiFM images of acceptor. (ii) PiFM images of the PM6. (iii) Combined images of (i) and (ii) to provide chemical map of PM6 and acceptor.

diffractions in the IP direction and π-π stacking (010) diffractions in the OOP direction. Their corresponding *d*-spacing and crystalline correlation lengths for the lamellar and π-π stacking peaks are summarized in Supplementary Fig. 24. Despite that all the blend films exhibit similar lamellar *d*-spacing (20.9–21.8 Å) and π-π *d*-spacing (3.63–3.71 Å), their corresponding crystal coherence length values of the TSMA-based blend films in both lamellar stacking (102.81 Å for PM6:TDY-α and 97.49 Å for PM6:TDY-β) and π-π stacking (25.70 Å for PM6:TDY-α and 24.91 Å for PM6:TDY-β, respectively) are clearly increased compared with those of PM6: Y6 blend films (with corresponding values of 73.44 and 24.80 Å).

To further examine the phase images of different blend films, we utilized an emergent technology, photo-induced force microscopy (PiFM)[58], by imaging at the characteristic Fourier transform infrared wavelengths corresponding to the absorption peaks of donor (1289 cm$^{-1}$) and acceptor (1697 cm$^{-1}$) species. As shown in Fig. 4e–g, the PiFM images demonstrate nm-scale patterns of the individual chemical components, exhibiting a unique BHJ bicontinuous-interpenetrating network with red color for the donor phase and green color for the acceptor phase. It can be inferred from Fig. 4e–g that the Y6-based blend film exhibits a smaller phase separation of around 11 nm, while the dimer-based films show a larger degree of phase separation, approximately 20 nm for TDY-α and approximately 24 nm for TDY-β. The larger phase separation of the dimers-based blend can also be

revealed by the TEM images (Supplementary Fig. 25). The morphology features of the TSMAs could also be associated with the relatively lower miscibility between TSMAs and PM6, which will be discussed in the next section. The oversized phase separation for the TDY-β-based blend may account for its poor device performance, while the more suitable domain size for the TDY-α-based blend, which is beneficial for exciton dissociation and charge transportation, can lead to higher photovoltaic performance.

**Improved device stability**

In addition to the better device efficiency, we noted that our tethered strategy can also help to enhance the stability, which is another essential advantage for such TSMAs. Herein, thermal stress is applied for unencapsulated devices under dark in nitrogen-filled glovebox. As shown in Fig. 5a, the TSMA-based device exhibits less burn-in loss of just 5% during the first 70 h, while that of the Y6-based device dropped by 15% within this timescale. Furthermore, the glass-encapsulated devices are aged in the air device under 1-Sun equivalent illumination from white LEDs at the maximum power point conditions in air, where continuously putting out current was calibrated by the measured *J–V* characteristic curves every hour. As seen in Fig. 5b, TDY-α based devices presented outstanding photo-stability, maintaining over 90% of initial PCE after operating for 200 h, and tended to be stable after the initial burn-in degradation. Ultimately, the photo-stability of TDY-α

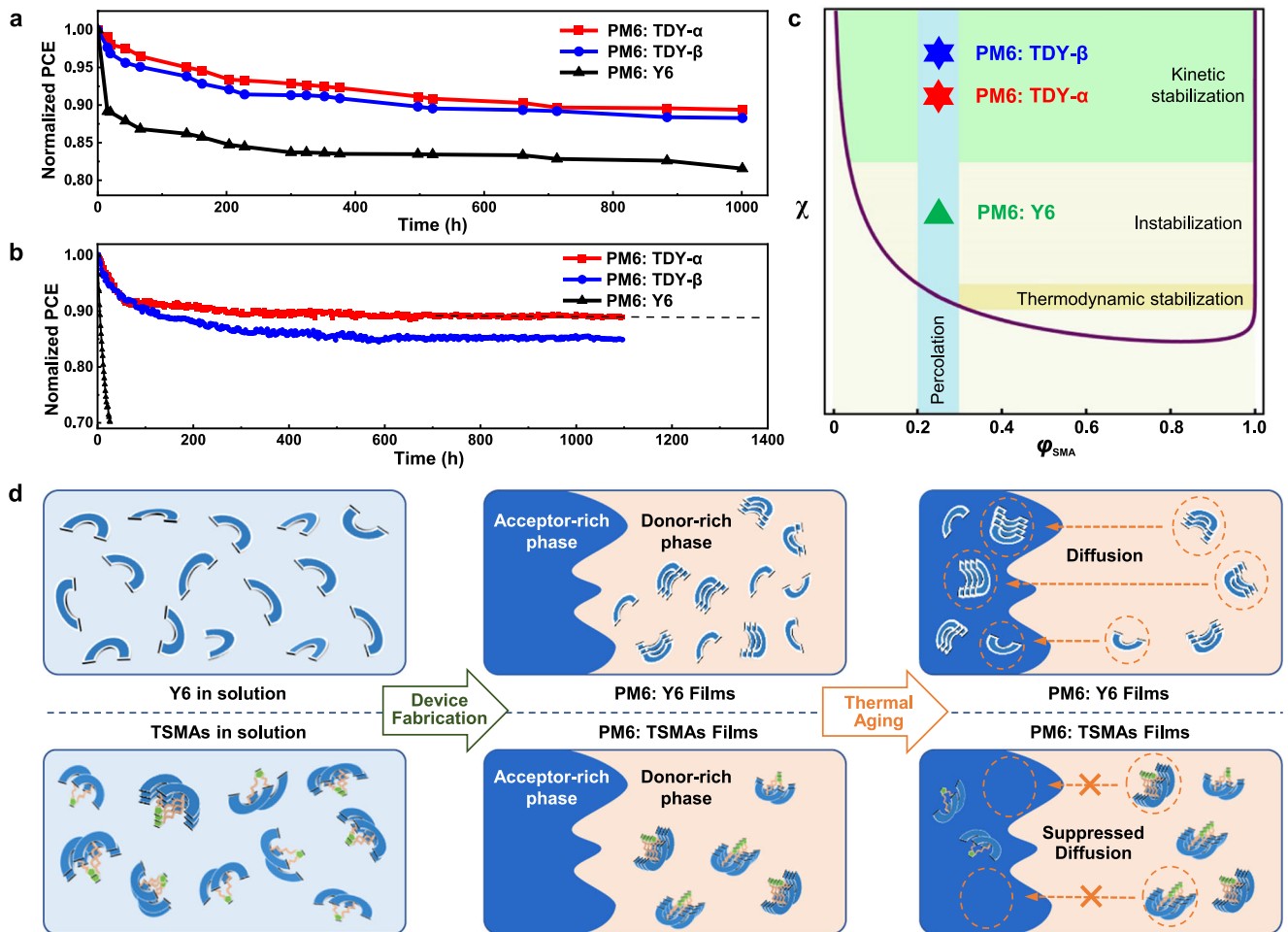

**Fig. 5 | Improved device stability. a** Normalized PCEs of the PM6:Y6 and PM6: TSMA-based devices under long-term annealing at 100 °C in a nitrogen-filled glovebox. **b** Maximum power point (MPP) stability test of the PM6:Y6 and PM6:TSMA-based devices under 1-Sun equivalent illumination from white LEDs at the MPP

conditions in open-air. **c** Illustrations of the dependence of the morphology stability on the χ value in the $\chi$-$\varphi_{SMA}$ phase diagram. **d** Schematic illustration of the TSMAs and Y6 molecular distribution and diffusion process in solution and PM6:acceptor films during the device fabrication and thermal aged procedures.

based devices is further conducted for over 1100 h, and the extrapolation of operational stability of TDY-α-based devices indicates an average $T_{80}$ lifetime of about 35,000 h (about 15 years, Supplementary Fig. 26) if operating average 7 h per day in actual open-air working condition at Beijing. Notably, it is also important to consider additional degradation pathways besides morphological stability[59]. The improved stability of TDY-α based devices may also be attributed to the reduced trap-assisted recombination that typically causes "burnin" degradation in devices. Regarding the photo-oxidation of our dimers, it is worth noting that they share the same Y6 subunits as Y6 itself. This suggests that they may be susceptible to similar light-induced damage. However, for the sake of simplicity, this factor was not considered in our analysis.

In the blend, the thermodynamic relaxation of acceptor domains or their nucleation will lead to changes in the microstructure in the bulk blend and coarsen surface. After thermal annealing of the blend films for 4 h, the change in the microstructure is clearly revealed by the evolution of their lamellar stacking CCLs values (Supplementary Fig. 27), exhibiting a remarkably increased tendency for the PM6: Y6 blend films relative to those of the TSMA-based blend. The evolution of the surface on a longer timescale also supports the stabilized morphology of the TSMA-based device. With the blend films annealed at 85 °C (Supplementary Fig. 28), for the Y6-based blend, an obviously sharp increase in its surface roughness was observed, suggesting the existence of larger scale phase separation. As expected, for the TSMA-

based blend, such an increase in surface roughness was effectively retarded, especially for that of TDY-α-based blend films. The result can be well explained by the larger $T_g$ values (115 °C for TDY-α, 106 °C for TDY-β) of the dimers relative to that of Y6 (99 °C), and suggest that the increase in the molecular size can retard the diffusion into the polymer domain. As the diffusion of SMAs into the donor polymer follows an Arrhenius behavior, thus the diffusion coefficients ($D_{85}$) at 85 °C can also be estimated based on their $T_g$ values through the equation proposed by Ade[60]. The calculated $D_{85}$ of PM6: Y6, PM6: TDY-α and PM6: TDY-β based blend films are $6.85 \times 10^{-18}$, $6.21 \times 10^{-19}$ and $2.40 \times 10^{-18}$ cm$^2$ s$^{-1}$, respectively. The $D_{85}$ of PM6: TDY-α-based blend is one order of magnitude lower than that of the PM6: Y6-based blend. The result suggested that PM6: TDY-α based blend is closer to the threshold of a kinetically stable system, demonstrating the effectiveness of our tethered strategy in suppressing the diffusion of acceptor molecules.

The high stability of the TSMA-based device can also be understood by the Flory–Huggins interaction parameter (χ) of different blend films, which is evaluated via the $T_m$ depression method of acceptors in homogeneous D:A mixtures with various D:A weight ratio (Supplementary Fig. 29)[16]. On the basis of Ade–O'Connor–Ghasemi theory, the morphology of PM6:Y6 blend is neither thermodynamically nor kinetically stabilized, and an appropriate increase of the χ value can enhance the morphological stability[19]. In our calculation, χ values of the PM6:TDY-α and PM6:TDY-β blends are 0.98,

1.09, respectively, both higher than that of PM6:Y6 blend (0.89). The result suggests a more hypo-miscible system for the dimer-based blend. Since a certain amount of mixed amorphous phases is crucial for efficient charge separation and extraction, a too-low miscibility (hypo-miscible, higher $\chi$) means over-purification of mixed domains, and a too-high miscibility (hyper-miscible, lower $\chi$) means insufficient phase separation, both of which will lead to performance deteriorations, thus, a proper D-A miscibility is necessary. The higher $\chi$ value for the dimer-based blends indicates a more hypo-miscible system compared to PM6:Y6 (Fig. 5c), resulting in suppressed diffusion-enabled demixing of the morphology. In a practical device exposed to light or heat, the morphological changes are controlled kinetically, making the $T_g$ a more plausible explanation for the difference in device stability.

To better understand the advantages of the tethered approach, the effect of their aggregation behavior on the morphology and thus device performance is schematically illustrated in Fig. 5d. Benefitted from the spacer, the TDY-$\alpha$ molecules pre-aggregate in solution, which can manipulate their aggregation behavior in their blend with polymer donor, affording better crystallinity and larger phase separation. Since the TSMAs exhibit relatively higher $\chi$ and $T_g$, which can significantly suppress the diffusion away from the polymer matrix to form a kinetically stabilized morphology based on a hypo-miscible status (Fig. 5c), achieving long-term stability and high efficiency.

## Discussion

Here two isomeric TSMAs based on different thiophene dicarboxylate-based spacers are designed and synthesized. The isomer effect provides the possibility to fully study the structure–property relationship with the variation on the molecular topology, affording TSMAs with different glass transition temperatures, crystallization behavior, and Flory–Huggins interaction parameter with the polymer donor. For the tethered dimers, they delivered folded geometries in solution, and the overlapping preference of individual Y6 subunit is related with the bent shape of the aromatic core, which triggers a different electronic coupling, thermodynamic property and aggregation behavior for the two dimers. TDY-$\alpha$ possesses a higher glass transition temperature, better crystallinity relative to its segment of Y6 and isomer counterpart of TDY-$\beta$, and a suitable Flory–Huggins interaction parameter with the polymer donor.

As a result, the TDY-$\alpha$-based devices exhibit a further increased efficiency as high as 18.1%, compared with previously reported TSMAs with a linear fashion. Most importantly, the TDY-$\alpha$-based devices possess substantially reduced burn-in efficiency loss relative to that of Y6 and its counterpart of TDY-$\beta$, retaining more than 80% of the initial PCE under long-term annealing at 80 °C for 1000 h or under continuous illumination for 1100 h. The extrapolation of operational stability of TDY-$\alpha$ based devices indicates an average $T_{80}$ lifetime of about 15 years if operating average 7 h per day in actual open-air working condition in Beijing. Our result suggests that with proper aromatic-core engineering, the TSMAs can achieve a high efficiency compared to the state-of-the-art SMAs, while showing more advantages of thermodynamic stability for future commercial applications. To fully understand the enhanced stability of the dimer-base device, it is also important to identify the potential impact of photodegradation on its stability under different conditions. While our research has primarily focused on the isomerization effect of the dimers, we also acknowledge the importance of the steric hindrance effect of the shoulder side chains on intermolecular packing. Further investigation of this effect could offer more opportunities to modulate the intermolecular packing of SMA subunits, leading to enhanced photophysical properties of the dimers. Related findings will pave the way for the development of more efficient and stable photovoltaic materials.

## Methods

### Reagents and materials

Polymer donor PM6 was purchased from Solarmer Materials. Y6 and 2-(5,6-difluoro-3-oxo-2,3-dihydro-1H-inden-1-ylidene)malononitrile (2F-IC) were purchased from eFlexPV. $C_{11}TT(N-H)BT$ (Y6 core) was purchased from Hyper, Inc. The other reagents and chemicals were purchased from Bidepharm Co. Ltd (Shanghai, China) and J&K Scientific Co. Ltd (Beijing, China) and were of analytical grade and used without further purification. The TSMAs were synthesized according to the chemical routes in Supplementary Fig. 1.

### Synthesis of compound 2a (or 2b)

The carboxylic acid derivatives 1a or 1b (2.00 mmol), 1,12-Dibromo-dodecane (10.00 mmol) and $K_2CO_3$ (10.00 mmol) were mixed in acetonitrile (50 ml) to stir at 60 °C for 24 h under argon atmosphere. After cooling to room temperature, the mixture was extracted with petroleum ether, and the solvent was removed, the organic layer was dried with $MgSO_4$ and the solvent was removed under vacuum. Compound 2a or 2b was obtained by column chromatography on silica gel using petroleum ether as eluent. The product was dried under vacuum to give 2a as a white solid with 82% yield and 2b as a colorless liquid with 75% yield. $^1H$ NMR of 2a (400 MHz, Chloroform-d) $\delta$ = 7.73 (s, 2H), 4.32 (t, 4H), 3.41 (t, 4H), 1.70-1.92 (m, 10H), 1.51-1.21 (m, 38H). $^1H$ NMR of 2b (400 MHz, Chloroform-d) $\delta$ = 7.85 (s, 2H), 4.28 (t, 4H), 3.43 (t, 4H), 1.67–1.93 (m, 10H), 1.52–1.22 (m, 38H).

### Synthesis of compound 3a (or 3b)

Compound 2a or 2b (0.1 mmol), $K_2CO_3$ (2 mmol), KI (1 mmol), Y6 core (0.22 mmol) were mixed in DMF (100 ml) to stir at 75 °C for 24 h under an argon atmosphere. After that, 2-ethylhexyl bromide (0.5 mmol) was added and the mixture was stirred for another 12 h. After cooling to room temperature, the mixture was extracted with dichloromethane. The organic layer was dried with $MgSO_4$ and the solvent was removed under vacuum. Compound 3a or 3b was obtained by column chromatography on silica gel using petroleum ether and dichloromethane as eluent. The product was dried under vacuum to give 3a as orange-red oil with 79% yield and 3b as red oil with 72% yield. $^1H$ NMR of 3a (400 MHz, Chloroform-d) $\delta$ = 7.72 (s, 2H), 7.02 (d, 4H), 4.55–4.70 (dd, 8H), 4.29 (t, 4H), 2.83 (t, 8H), 2.09 (dt, 2H), 1.95–0.60 (m, 152H). $^1H$ NMR of 3b (400 MHz, Chloroform-d) $\delta$ = 7.82 (s, 2H), 7.02 (d, 4H), 4.55–4.70 (dd, 8H), 4.25 (t, 4H), 2.83 (t, 8H), 2.09 (dt, 2H), 1.95–0.60 (m, 152H).

### Synthesis of compound 4a (or 4b)

Phosphorus oxychloride (0.5 ml) was added by the injector to DMF (1 ml) in a two-necked flask, and the mixture was stirred for 30 min at 0 °C under an argon atmosphere. The mixture was then transferred to a solution of compound 3a or 3b (0.1 mmol) with 50 ml dichloroethane in another two-necked flask at 0 °C, and refluxed at 90 °C overnight under an argon atmosphere. The mixture was washed with a saturated aqueous solution of sodium carbonate and extracted by dichloromethane. The organic layer dried with $MgSO_4$ and the solvent was removed under vacuum. Compound 4a or 4b was obtained by column chromatography in a silica gel column using dichloromethane as an eluent. The product was dried under vacuum to give 4a or 4b as red solid (89% yield, 85% yield). $^1H$ NMR of 4a (400 MHz, Chloroform-d) $\delta$ = 10.15 (s, 4H), 7.71 (s, 2H), 4.57–4.76 (dd, 8H), 4.28 (t, 4H), 3.20 (t, 8H), 2.07 (dt, 2H), 1.95–0.60 (m, 152H). $^1H$ NMR of 4b (400 MHz, Chloroform-d) $\delta$ = 10.14 (s, 4H), 7.82 (s, 2H), 4.58–4.73 (dd, 8H), 4.24 (t, 4H), 3.20 (t, 8H), 2.06 (dt, 2H), 1.95–0.60 (m, 152H).

### Synthesis of TDY-$\alpha$ (or TDY-$\beta$)

Compounds 4a or 4b (0.10 mmol) and 2F-IC (0.45 mmol) were dissolved in toluene (5 ml). $BF_3 \cdot OEt_2$ (2.25 mmol) and acetic anhydride (0.1 ml) were added, and the reaction mixture was stirred at room

temperature for 15 min. Then, the reaction mixture was added dropwise into methanol with stirring. The precipitate was collected, washed with methanol several times, then dissolved in chloroform and added dropwise into methanol again to afford crude product. This process was repeated several times until the target SMAs were pure enough for photovoltaic devices. $^1$H NMR of TDY-α (400 MHz, Chloroform-d) δ = 8.79 (s, 2H), 8.59–8.33 (m, 6H), 7.83 (d, 2H), 7.56 (dt, 4H), 4.57 (d, 8H), 4.37 (t, 4H), 2.89 (s, 8H), 2.50 (s, 4H), 2.03–0.50 (m, 150H). $^{13}$C NMR of TDY-α (100 MHz, CDCl3, δ): 185.80, 185.67, 161.48, 157.87, 157.15, 155.34, 153.68, 153.47, 152.87, 147.18, 146.75, 145.03, 144.99, 138.59, 136.75, 136.67, 136.47, 136.12, 134.72, 134.21, 134.06, 133.91, 133.56, 133.49, 133.19, 132.77, 132.35, 132.02, 130.72, 130.04, 119.69, 119.30, 114.86, 114.64, 114.44, 113.33, 112.74, 112.23, 112.04, 111.56, 77.21, 69.12, 69.02, 65.75, 54.93, 51.63, 40.68, 31.94, 31.92, 31.83, 30.94, 30.76, 30.12, 30.08, 29.94, 29.89, 29.85, 29.73, 29.68, 29.64, 29.51, 29.50, 29.47, 29.40, 29.36, 29.34, 28.59, 28.02, 27.43, 26.42, 23.38, 22.75, 22.70, 14.13, 13.71, 10.35. MS (MALDI-TOF) of TDY-α $m/z$: [M+H]$^+$ calcd for C$_{178}$H$_{188}$F$_8$N$_{16}$O$_8$S$_{11}$, 3182.16, found: 3182.007. $^1$H NMR of TDY-β (400 MHz, Chloroform-d) δ = 8.72 (s, 2H), 8.53–8.39 (m, 6H), 7.89 (d, 2H), 7.54 (dt, 4H), 4.55 (d, 8H), 4.34 (t, 4H), 2.86 (s, 8H), 2.58 (s, 4H), 2.03–0.50 (m, 150H). $^{13}$C NMR of TDY-β (100 MHz, CDCl3, δ): 185.70, 163.55, 157.73, 157.14, 155.40, 155.26, 153.59, 153.43, 152.95, 152.81, 152.67, 147.15, 146.71, 145.04, 144.95, 136.70, 136.48, 136.20, 134.54, 133.94, 133.62, 133.49, 132.65, 132.30, 131.93, 131.49, 130.81, 130.04, 119.57, 119.33, 114.84, 114.62, 114.40, 113.27, 112.63, 112.15, 111.95, 111.81, 111.63, 77.22, 69.18, 69.11, 65.52, 54.81, 51.62, 40.69, 31.94, 31.92, 30.90, 30.77, 30.56, 30.30, 30.15, 30.03, 29.88, 29.83, 29.76, 29.72, 29.69, 29.64, 29.62, 29.50, 29.49, 29.38, 29.36, 29.34, 28.74, 28.06, 27.67, 26.08, 23.41, 22.71, 14.13, 13.71, 10.34. MS (MALDI-TOF) of TDY-β $m/z$: [M+H]$^+$ calcd for C$_{178}$H$_{188}$F$_8$N$_{16}$O$_8$S$_{11}$, 3182.16, found: 3181.809.

## Material characterization

For the NMR tests, $^1$H NMR and $^{13}$C NMR were recorded on Bruker AVANCE 400 MHz NMR spectrometer with CDCl$_3$ as solvent. The $^1$H-$^1$H NOESY NMR spectra were recorded on a JNM-ECZ600R/S3 (Jeol, Japan) (600 MHz). MALDI-TOF mass spectrometry experiments were performed on an autoflex III instrument (Bruker Daltonics, Inc.). TGA was measured on HTG-1 Thermogravimetric Analyzer (Beijing Hengjiu Experiment Equipment Co. Ltd.) with a heating rate of 10 °C min$^{-1}$ under a nitrogen flow rate of 100 ml min$^{-1}$. DSC measurements were performed on a Mettler Toledo DSC1 star system, the heating rate and cooling rate were both kept 10 °C min$^{-1}$ under a nitrogen flow rate of 75 ml min$^{-1}$. The samples were loaded in aluminum pans directly with another empty aluminum pan as the reference. As for the blend samples, donor and acceptor materials were solved in chloroform (20 mg ml$^{-1}$ for acceptor, gradient proportion of donor by mass concentration) and stirred overnight. Next, the solution was spin-coated onto cleaned glass substrates and dried under a vacuum to form homogeneous films. The samples were then scraped off the substrates and loaded in aluminum pans. Due to the thermal decomposition of the dimers over 280 °C, we can only track the DSC signal below that temperature. So the test ended at 270 °C, from which the whole melting peak was detected. In our test, unfortunately, no liquid crystal phase was observed. In our DFTB calculation, for the dimer with a linear fashion, its energy was set to zero. Based on this, the relative energy was provided for different geometries. Cyclic voltammetry was conducted on a Zahner IM6e electrochemical workstation using sample films coated on glassy carbon as the working electrode, Pt wire as the counter electrode, and Ag/AgCl as the reference electrode, in a 0.1 M tetrabutylammonium hexafluorophosphate (Bu$_4$NPF$_6$) acetonitrile solution and ferrocene/ferrocenium (Fc/Fc$^+$) couple was used as an internal reference. From the onset oxidation potential ($\varphi_{ox}$) and onset reduction potential ($\varphi_{red}$), the highest occupied molecular orbital (HOMO) energy level ($E_{HOMO}$) and the lowest unoccupied molecular orbital (LUMO) energy level ($E_{LUMO}$) of the acceptors are

calculated according to the equation of $E_{LUMO/HOMO} = -e\ (\varphi_{red/ox} + 4.8 - \varphi_{Fc/Fc+})$ (eV), where the unit of $\varphi_{red/ox}$ is V vs. Ag/AgCl and $\varphi_{Fc/Fc+}$ is 0.434 V vs. Ag/AgCl in our measurement system. The UV-vis absorption spectra were measured by a Hitachi U-2910 UV-vis spectrophotometer. In the case of solution absorbance measurement, the dilute solution of acceptors in chloroform ($1 \times 10^{-5}$ M) was prepared to be measured. Besides, the thin film samples were prepared by spin-coating (3000 rpm) chloroform solution (10 mg ml$^{-1}$) of acceptors on quartz plates. The as-cast thin films all performed a thickness ranging from 50 to 80 nm, which were recorded on the Bruker DEKTAK XT step profiler. Absorption spectra of acceptors at various annealing temperatures were measured ex situ to fit their $T_g$. After spin-coating, the thin films were annealed for 5 min in air at various temperatures (25 or 160 °C) depending on their nominal $T_g$. AFM measurements were performed using a Dimension Icon2-SYS AFM instrument (Bruker) in the tapping mode. The GIWAXS measurements were conducted at PLS-II 6A U-SAXS beamline of the Pohang Accelerator Laboratory in Korea.

## The Flory–Huggins interaction parameters (χ)

These parameters of different blend films are evaluated via the $T_m$ depression method of acceptors in homogeneous D:A mixtures with various D:A weight ratio. The related calculation equation was developed by Nishi and Wang[60–63], as shown below:

$$\frac{1}{T_m} - \frac{1}{T_m^0} = -\frac{Rv_2}{v_1 \Delta H_f} \left[ \frac{\ln\varphi_2}{m_2} + \left( \frac{1}{m_2} - \frac{1}{m_1} \right)(1 - \varphi_2) + \chi(1 - \varphi_2)^2 \right] \quad (1)$$

In Eq. (1), subscripts 1 and 2 represent amorphous donor material and crystalline acceptor material, respectively; $T_m$ and $T_m^0$ are the melting points of the D:A mixtures and the pure crystalline acceptors; $\Delta H_f$ represents the heat of fusion of the crystalline acceptors; $R$ is the ideal gas constant; $v_1$ and $v_2$ represent the molar volumes; $m$ is the degree of polymerization; and $\varphi$ is the volume fraction. In this work, subscripts 1 and 2 represent PM6 and SMAs, respectively. For the PM6:acceptors mixtures, since the degree of polymerization of PM6 is over large compared to acceptors, $m_1$ can be seen as ∞ and $m_2$ to be 1, so that Eq. (1) can be simplified as follows:

$$\frac{1}{T_m} - \frac{1}{T_m^0} = -\frac{Rv_2}{v_1 \Delta H_f} \left[ \ln\varphi_2 + (1 - \varphi_2) + \chi(1 - \varphi_2)^2 \right] \quad (2)$$

Moreover, $\chi$ can be of the following form if neglect the effects of entropy and $\varphi_2$,

$$\chi = \frac{\beta v_1}{RT_m} \quad (3)$$

where $\beta$ represents the interaction energy density characteristic of the organic material pair. By substituting Eq. (3) into Eq. (2), Eq. (4) is obtained as below, and the linear relationship between $-[1/T_m - 1/T_m^0 + Rv_2(\varphi_1 + \ln\varphi_2)/(\varphi_1 \Delta H_f)]/\varphi_1$ and $\varphi_1/T_m$ represents the corresponding $\chi$ values.

$$-\frac{1}{\varphi_1} \left[ \frac{1}{T_m} - \frac{1}{T_m^0} + \frac{Rv_2}{v_1 \Delta H_f}(\varphi_1 + \ln\varphi_2) \right] = \frac{\beta v_2}{\Delta H_f} \cdot \frac{\varphi_1}{T_m} \quad (4)$$

## Device fabrication

The conventional device structure of ITO/PEDOT:PSS/active layer/PDINN/Ag was constructed. The indium tin oxide (ITO) substrates were prepared in an order of deionized water, acetone, ethanol, then dried in oven at 100 °C for 30 min. The substrates were treated with ultraviolet ozone for 10 min and the PEDOT: PSS aqueous solution (Baytron P 4083 from H. C. Starck) was filtered through a 0.45-mm filter and then spin-coated on precleaned ITO-coated glass at 6000 rpm for 30 s.

After annealing at 150 °C on hot plate for 20 min, the substrates were transferred into an N$_2$-protected glovebox. All the active layers were obtained by spin-coating the chloroform solution containing D:A blend (w/w, 1:1.2) in a total concentration of 16 mg ml$^{-1}$. Subsequently, -10 nm PDINN as cathode interlayer was spin-coated onto the active layers in a concentration of 1 mg ml$^{-1}$ in methanol solution. Finally, about 100 nm Ag was vacuum thermally deposited on the top of the device through a shadow mask. For the thermal stressed device, to avoid the diffusion of the organic cathode interlayer, inverted devices were fabricated with a device structure of glass/ITO/ZnO (20 nm)/PM6: acceptors (100 nm)/MoO$_x$(10 nm)/Ag(100 nm). Subsequentially, 10 nm MoO$_x$ and 100 nm Ag were thermally evaporated with a shadow mask on the top of the active layer.

## Device characterization

The current–density–voltage (*J–V*) characteristics were measured by using the solar simulator (SS-F5-3A, Enlitech, Taiwan) along with AM 1.5G (100 mW cm$^{-2}$). The EQE was recorded with a QE-R measurement system (Enlitech, Taiwan). The effective area of all devices was confined to 0.06 cm$^2$. The hole-only and electron-only devices were fabricated with the architectures of ITO/PEDOT:PSS/active layer/MoO3/Ag and ITO/ZnO/active layer/PDINN/Ag, respectively. Hole-only and electron-only devices were recorded with a Keithley 236 sourcemeter under dark. The hole and electron mobility were determined by fitting the dark current to the model of single-carrier SCLC, which is described by Eq. (5),

$$J = \frac{9}{8}\varepsilon_0\varepsilon_r\mu\frac{V^2}{d^3} \quad (5)$$

where *J* is the current–density, *μ* is the zero-field mobility, *ε$_O$* is the permittivity of free space, *ε$_r$* is the relative permittivity of the material, *d* is the thickness of the active layers, and *V* is the effective voltage. The effective voltage was obtained by subtracting the built-in voltage (*V$_{bi}$*) and the voltage drop (*V$_s$*) from the series resistance of the whole device except for the active layers from the applied voltage (*V$_{appl}$*), *V* = *V$_{appl}$* − *V$_{bi}$* − *V$_s$*. The hole and electron mobilities can be calculated from the slope of the *J$^{1/2}$* – *V* curves.

## Degradation of devices under white light

The long-term stability of encapsulated devices was evaluated using a multi-channel solar cell performance decay test system (PVLT-6001M-32A, Suzhou D&R Instruments Co. Ltd.). In our test, the glass-encapsulated devices were exposed to continuous white LED light (D&R Light, L-W5300KA-150, Suzhou D&R Instruments Co. Ltd.) while being stored in the air. The initial illumination intensity was adjusted to match the J$_{SC}$ measured under standard conditions by AM 1.5G. During the test, the Illumination intensity was monitored using a photodiode. Periodic *J–V* characterization of the devices allowed for the calculation of photovoltaic parameters, including *V$_{OC}$*, *J$_{SC}$*, FF, and PCE according to the *J–V* curves.

## Reporting summary

Further information on research design is available in the Nature Portfolio Reporting Summary linked to this article.

## Data availability

The data that support the findings of this study are available from the corresponding author on request.

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

## Acknowledgements

The work was supported by the National Natural Science Foundation of China (Nos. 21734008, 22175014, and 22273062), Fundamental Research Funds for the Central Universities (buctrc201822). Q.Z. thanks the NanHu Young Scholar Supporting Program of XYNU. C.Y. thanks the National Research Foundation of Korea (NRF) grant funded by the Korean government (MSIP) (2021R1A2C3004202).

## Author contributions

Z.-G.Z and Y.L. supervised the project. Z.-G.Z, and Y.L. contributed to the results analysis of all the experiments. Z.-G.Z. and Y.B. designed the tethered acceptors, Y.B. synthesized them, Ce.Z. and B.C. repeated the

experiments, L.X., S.L. and H.F. optimized the conditions for synthesis. Z.Z. carried out the device characterization, analyzed the device parameters, and measured the AFM. Q.Z. measured the 1H-1H NOESY NMR spectra. Y.B. measured the UV, CV, TGA, DSC, and H.W. assisted in the measurement of TGA and DSC. W.C. and W.L. measured the TEM. Y.B. and Q.C. calculated the $\chi$ values and analyzed the results. Y.Z., C.Y., S.K., Ch.Z., Y.O., R.Z. and F.G. measured the GIWAXS, and L.Y., M.G. analyzed the results. H.G. performed the conformation calculation of the TSMAs. Y.B. and Z.-G.Z., wrote the paper, and Z.Z., Y.L. revised the manuscript. All authors discussed the results and commented on the final manuscript.

## Competing interests

The authors declare no competing interests.
