## [Peer Review File · Nature Communications]

Geometry Design of Tethered Small-molecule Acceptor towards High Stable and Efficiency Polymer Solar CellsREVIEWER COMMENTS

Reviewer #1 (Remarks to the Author):

Dimerization of small molecular acceptors (SMA) recently emerged to be a promising strategy to address the thermal instability of monomeric SMAs in its blend with polymer donor, which is a critical issue for the real application of organic solar cell (OSCs). However, the device efficiency of those dimeric SMAs is still inferior to those of state-of-the-art SMAs (such as Y6). Here, the authors manipulated the geometry of tethered SMAs via an isomeric engineering on the aromatic-core, which can simultaneously achieve higher PCEs (over 18.0%) and ultra-long life-time stability (an extrapolated T80 lifetime of about 35000 h). Besides, a clear structure-property relationship was provided via the investigation of the geometric effect, thus providing a rational design principle for high-performance dimeric SMAs. Interestingly, a folded geometry of the tethered SMAs was found in solution, which is fundamentally important to better understand the aggregation behavior in films. Their result reasoned that with a proper geometry design, the tethered SMAs can achieve a high efficiency comparable to that of the state-of-the-art SMAs, while showing more advantages of operating stability. Thus, I am pleased to recommend the acceptance of this manuscript in Nature Communication after well addressing the following issues.

1. For the 2D 1H-1H NMR spectra, NOESY and ROESY are close related with the intermolecular interactions. What about the result from the 2D 1H-1H ROESY NMR? Is the result consistent with that obtained from the NOESY?
2. The two tethered dimers, TDY- α and TDY- β , are isomers bearing different aromatic-core. The reviewer is interested to know their film molar absorptivity, is there any difference?
3. The calculation of the total energy for the two dimers confirmed a folded conformation for the two dimers, which is consistent with the 2D 1H-1H NMR spectra. However, the result was not well presented in the manuscript. It is suggested that the authors reorganize their discussion on the results.
4. The 2D 1H-1H NMR spectra confirms the existence of folded geometry, which shows a lower total energy as suggest by their calculation. Thus, the folded geometry is the dominated conformation, but that doesn't mean the absence of any un-folded one. It is suggested that Figure 3a give the conformational equilibrium between folded and un-folded conformation, and the former is the dominated one.
5. TDY- β shows a larger χ value, while a worse device stability compared to that of TDY- α . Comment should be given for this phenomenon.
6. The folded conformation of the dimers is an important finding in this work, which should be highlighted in the conclusion part.

Reviewer #2 (Remarks to the Author):

Although numerous new non-fullerene acceptors (NFA) were developed in the last few years, high-performance, highly stable non-fullerene acceptors are still highly needed. This manuscript reports the design and synthesis of two tethered dimeric Y6 derivatives with 2,5-(TDY- α) and 3,4-thieno (TDY- β) core units, which showed excellent photovoltaic performance and stability. The structure-properties-performance relationship was well investigated, providing a solid conclusion that the topology of the well-known NFA molecule with proper molecular structure design could be an effective way to create ideal acceptor molecules for organic photovoltaics. The manuscript was well prepared, and it is a pleasure to read this paper. I would highly recommend accepting this manuscript for publishing in Nat. Commun. after the following revision suggestion:

1. Figure 1g and 1f. The calculated total energies of TDYs are quite different. The TDY- α showed much lower total energy than TDY- β , which means TDY- α has a high tendency to aggregate. This is counterintuitive since the angle between the two carbonyl groups of TDY- β is smaller than TDY- α (Figure 1a) and closer for the Y6 moieties in TDY- β (Figures 1i and 1j). Could the authors provide a more detailed explanation on this result? Also, I would suggest the authors to keep the same X-axis identical for better comparison.
2. Interestingly, both TDYs showed additional absorption band at 693 nm, which was ascribed to “the electronic state coupling of individual SMA in one dimer molecule” (Page 7, line 147-150). I fully agree with the authors on this. However, could the authors confirm whether this is due to the formation of H-aggregates of Y6 moiety in solution?
3. About the GIWAXS results. In figure 2e and 2h, shoulder peaks were measured both in TDY- α and TDY- β (an additional broad ring in Figure 1e). What could this shoulder peak represent? Interestingly, such a shoulder peak was found in PM6:Y6 blend film (Figure 4c), but not in PM6:TDYs blends (Figure 4a and 4b). Could the authors confirm whether this is correct? And could the authors provide a detailed explanation on these?
4. About DSC results, the DSC traces of TDYs are limited to 270 °C. What about the full DSC trace of TDYs till melting? Could we see any liquid crystalline phases of TDYs?
5. About the HOMO/LUMO energy levels. It is known that the electronic interaction of two conjugated moieties would generally lead to higher HOMO and lower LUMO and a lower optical band gap. However, in TDYs, increased band gaps were measured with higher LUMO energy levels. What could be the reason for this?
6. The stability results are very impressive. Could the authors provide the instrument information for the photostability measurement? As for estimating the cells' lifetime in year, I would suggest the authors to check the average light hours in Beijing. With this, more accurate life years would be obtained.

Reviewer #3 (Remarks to the Author):

Zhang et al. synthesized two types of tethered dimeric small-molecular acceptors (TDY- α , TDY- β) and studied their photovoltaic properties depending on the tethering angles. Authors reported TDY- α has a higher glass transition temperature, better crystallinity relative to its individual Y6 segment and isomer counterpart of TDY- β , and a more stable morphology with the polymer donor. As a result, the TDY- α based PSCs delivered a better PCE as high as 18.1%, and T80 lifetime of 35000 h. However, the intra- and intermolecular interactions of TDY- α and TDY- β are not clear with ambiguous characterizations including 2D NMR and GIWAXS, etc. I doubt the suggested mechanism and discussions for the higher PCE and operational stability of TDY- α -based devices.

- Authors discussed the intra- and intermolecular interactions of TDY- α and TDY- β based on the 2D NOESY NMR and DFTB calculations, suggesting cofacial packing of Y6 moieties depending on the tethering angles. Although several 2D NMRs are presented in Supporting Information, the 2D NMR analysis is too ambiguous and it is hard to understand the different packing interactions of TDY- α and TDY- β molecules. In this paper, the different PCEs and stability were interpreted in terms of the different packing interactions of TDY- α and TDY- β , which should be analyzed clearly and carefully. In addition, DFTB calculation shows a cofacial interaction of two Y6 moieties in the tethered dimeric structures. However, Y6 molecules were reported to have a terminal-terminal packing interaction (between the neighboring INCN moieties) based on the XRD analysis. The core-core interaction is hindered because of the steric repulsion of undecyl side chains in the Y6 core. Authors did not consider the steric hindrance of core-core packing in their calculation, which may yield a wrong conclusion. Furthermore, the GIWAXS data of pristine films of three molecules are very similar and I cannot find the different scattering peaks which is related to the cofacial interaction of two Y6 moieties in the tethered dimeric structures which authors emphasized.

- Authors discussed the higher long-term thermal and light soaking stability of PM6:TDY compared to PM6:Y6. What is the origin of even worse light soaking stability of PM6:Y6? What is the origin of the higher photostability of dimeric TDY structures?

- Authors claimed the higher crystallinity of TDY- α and TDY- β compared to Y6. However, Y6 shows the highest heat of melting (28.15 J/g) compared to the dimeric structures (13.4 and 8.7 for TDY- α and TDY- β) according to Table 1. Actually, Y6 shows the highest crystalline interaction. The CCL values based on the 010 peak in the z direction are also contradictory. Both data are not consistent without supporting the authors' claims.

- "TSMAs feature a much stronger 0-1 absorption peaks at around 693 nm. The obvious difference can also be observed in the UV-vis absorption spectra even in ultra-dilute solution (Figure 2c, Figure S9), suggesting that there may exist some intrinsic aggregation in dilute solution for the dimers." This discussion is also very ambiguous. Compared to Y6, two dimeric structures show the higher 0-1, 0-2, 0-3 peaks in dilute solutions. Another interpretation is that these stronger vibronic peaks may be related to the different oscillation strength. The data analysis is too rough and ambiguous.

- What is the origin of different decomposition temperatures of the isomeric TDY- α and TDY- β ?

- Authors discussed that “Based on the AFM results, the TSMAs-based devices show relatively larger Rq (roughness of root mean square deviation) in the height images and more homogeneous phase separation in the phase images, suggesting an enhanced aggregation characteristic for the TSMAs and larger degree of phase separation. The TEM images also show a larger domain size for the TSMAs-based blend, fully consistent with that from the TEM images.” This discussion is not consistent with the data in Figure 4. The PM6:Y6 AFM phase image shows the clearly larger phase separation than the other two blends. In addition, TSMAs-based devices show relatively larger Rq (roughness of root mean square deviation) in the height images and more homogeneous phase separation are contradictory.

Response to Reviewer #1:

Dimerization of small molecular acceptors (SMA) recently emerged to be a promising strategy to address the thermal instability of monomeric SMAs in its blend with polymer donor, which is a critical issue for the real application of organic solar cell (OSCs). However, the device efficiency of those dimeric SMAs is still inferior to those of state-of-the-art SMAs (such as Y6). Here, the authors manipulated the geometry of tethered SMAs via an isomeric engineering on the aromatic-core, which can simultaneously achieve higher PCEs (over 18.0%) and ultra-long life-time stability (an extrapolated T80 lifetime of about 35000 h). Besides, a clear structure-property relationship was provided via the investigation of the geometric effect, thus providing a rational design principle for high-performance dimeric SMAs. Interestingly, a folded geometry of the tethered SMAs was found in solution, which is fundamentally important to better understand the aggregation behavior in films. Their result reasoned that with a proper geometry design, the tethered SMAs can achieve a high efficiency comparable to that of the state-of-the-art SMAs, while showing more advantages of operating stability. Thus, I am pleased to recommend the acceptance of this manuscript in Nature Communication after well addressing the following issues.

1) *For the 2D 1H-1H NMR spectra, NOESY and ROESY are close related with the intermolecular interactions. What about the result from the 2D ¹H-¹H ROESY NMR? Is the result consistent with that obtained from the NOESY?*

Response: Thanks for the reviewer's insightful comments. For the dimers, we carefully characterized both the NOESY and ROESY spectra to study the intermolecular interactions. In the revised manuscript, we provided the ROESY spectra in Figure S1 in the supporting information, and double checked our results. We found the result obtained from NOESY also support the folded geometry, consistent with that from ROESY. As the off-diagonal signals from ROESY spectra was too weak, we only chose the NOESY spectra in the main text to discuss the intermolecular interactions for clarity.

2) *The two tethered dimers, TDY- α and TDY- β , are isomers bearing different aromatic-core. The reviewer is interested to know their film molar absorptivity, is there any difference?*

Response: Following the suggestion, we characterized the film molar absorptivity of TDY- α and TDY- β in Figure R1. Related discussion was added in Page 9 of the revised manuscript. **“The film absorption coefficient was measured, showing high values of $1.15 \times 10^5 \text{ cm}^{-1}$ for TDY- α or $1.12 \times 10^5 \text{ cm}^{-1}$ for TDY- β .”**

Figure R1. Film molar absorptivity TDY- α and TDY- β .

3) *The calculation of the total energy for the two dimers confirmed a folded conformation for the two dimers, which is consistent with the 2D ^1H - ^1H NMR spectra. However, the result was not well presented in the manuscript. It is suggested that the authors reorganize their discussion on the results.*

Response: According to the advice, we reorganize the discussion in Page 7:

“Generally, the electrostatic attractions between electron-rich D-units and electron-deficient A-units result in the overlapping of individual Y6 segment, thus delivering a folded geometry to refine the total energy in a dimer. Interestingly, there are essentially lowest total energy geometric structure for both dimers: revealing 32.7° tilt angle for **TDY- α** system and 41.5° tilt angle for **TDY- β** . With a refined geometry, the top view and side view of the chemical structures of **TDY- α** are respective provided in **Figure 1h** and **1i**, while those of **TDY- β** are respective provided in **Figure 1k** and **1l**.”.

4) *The 2D ^1H - ^1H NMR spectra confirm the existence of folded geometry, which shows a lower total energy as suggest by their calculation. Thus, the folded geometry is the dominated conformation, but that doesn't mean the absence of any un-folded one. It is suggested that Figure 3a give the conformational equilibrium between folded and un-folded conformation, and the former is the dominated one.*

Response: According to the advice, we have added the illustration of the conformational equilibrium between folded and un-folded conformation, as shown in Figure 1c.

5) *TDY- β shows a larger χ value, while a worse device stability compared to that of TDY- α . Comment should be given for this phenomenon.*

Response: Thanks for the insightful comments. As we known, the Flory-Huggins interaction parameter χ value is a thermodynamically parameter revealing the terminal state of a blend film (Joule,

2021, 5, 2129). Due to the larger χ value of the dimers, thus we can conclude the dimers have a potentially better stability than that of Y6. While for a real device under light or heat, since the morphology changing is a dynamically controlled process, thus, the kinetic dynamic parameter, especially, the glass transition temperature (T_g) is more plausible to explain the device stability. Due to a lower T_g value of TDY- β (106°C) relative to that of TDY- α (115°C), it gives a worse device stability compared to that of TDY- α . Moreover, the slightly better stability of the TDY- α based devices is probably also related to the suppressed trap-assisted recombination, which usually induces so-called “burn-in” degradation in devices.

6) *The folded conformation of the dimers is an important finding in this work, which should be highlighted in the conclusion part.*

Response: Thanks for your advice. The folded conformation of the dimers was highlighted in the conclusion part (Page 18) of the revised manuscript, “For the tethered dimers, they delivered a folded geometries in solution, and the overlapping preference of individual Y6 segment are related with the bent shape of the aromatic core, which triggers a different electronic coupling and aggregation behavior for the two dimers both in solution and films. TDY- α possesses a higher glass transition temperature, better crystallinity relative to its segment of Y6 and isomer counterpart of TDY- β , and a suitable Flory-Huggins interaction parameter with the polymer donor.”

Response to Reviewer #2:

Although numerous new non-fullerene acceptors (NFA) were developed in the last few years, high-performance, highly stable non-fullerene acceptors are still highly needed. This manuscript reports the design and synthesis of two tethered dimeric Y6 derivatives with 2,5-(TDY- α) and 3,4-thieno (TDY- β) core units, which showed excellent photovoltaic performance and stability. The structure-properties-performance relationship was well investigated, providing a solid conclusion that the topology of the well-known NFA molecule with proper molecular structure design could be an effective way to create ideal acceptor molecules for organic photovoltaics. The manuscript was well prepared, and it is a pleasure to read this paper. I would highly recommend accepting this manuscript for publishing in Nat. Commun. after the following revision suggestion:

1) *Figure 1g and 1f. The calculated total energies of TYDs are quite different. The TDY- α showed much lower total energy than TDY- β , which means TDY- α has a high tendency to aggregate. This is counterintuitive since the angle between the two carbonyl groups of TDY- β is smaller than TDY- α (Figure 1a) and closer for the Y6 moieties in TDY- β (Figures 1i and 1j). Could the authors provide a*

more detailed explanation on this result? Also, I would suggest the authors to keep the same X-axis identical for better comparison.

Response: We strongly agreed with the reviewers that TDY- β is easier to form a folded geometry under a bent fashion. In our calculation, for the dimer with a linear fashion, its energy was set to zero. Base on which, the relative energy was provided for different geometries. So the direct comparison of the lowest energy states between TDY- α and TDY- β is meaningless. In our calculation, for TDY- α , the lower total energy just means a larger energy barrier between the un-folded geometry and folded geometry. Our result confirmed that TDY- β is more easily folded. Besides, following the suggestion, the X-axis have adjusted to keep the units as the same in the revised Figures 1g and 1j.

2) Interestingly, both TDYs showed additional absorption band at 693 nm, which was ascribed to “the electronic state coupling of individual SMA in one dimer molecule” (Page 7, line 147-150). I fully agree with the authors on this. However, could the authors confirm whether this is due to the formation of H-aggregates of Y6 moiety in solution?

Response: We totally agree with the reviewer that peak at 690 nm is originated from the stacking behavior of the Y6 subunits, more especially H-like aggregation of the dimers in solution. Also expanded explain for such peaks was added in Page 8 of the revised manuscript.

“When the two Y6-subunits folded, their electronic states become coupled, leading to the Davydov splitting⁵¹ of the original energy level for a new absorption band at 690 nm. The strength of the coupling between the electronic states depends on the distance and orientation of the two Y6 subunits, thus their strength is different between them.^{51, 52}”

3) About the GIWAXS results. In figure 2e and 2h, shoulder peaks were measured both in TDY- α and TDY- β (an additional broad ring in Figure 1e). What could this shoulder peak represent? Interestingly, such a shoulder peak was found in PM6:Y6 blend film (Figure 4c), but not in PM6:TDYs blends (Figure 4a and 4b). Could the authors confirm whether this is correct? And could the authors provide a detailed explanation on these?

Response: Thanks for your reminder. We double checked our GIWAXS results and line-cuts that take from GIWAXS. It can be seen that there are no such signal in the line-cuts, so we turn our attention to the data processing flow, and found the “shoulder peak” originated from the background. To remove such misunderstandings, this was corrected in Figure 2f and 4b in our revised manuscript.

4) *About DSC results, the DSC traces of TDYs are limited to 270 °C. What about the full DSC trace of TDYs till melting? Could we see any liquid crystalline phases of TDYs?*

Response: Due to the thermal decomposition of the dimers over 280°C, we can only track the DSC signal below that temperature. So the test was ended at 270°C, from which the whole melting peak was detected. In our test, unfortunately, no liquid crystal phase was observed.

5) *About the HOMO/LUMO energy levels. It is known that the electronic interaction of two conjugated moieties would generally lead to higher HOMO and lower LUMO and a lower optical band gap. However, in TDYs, increased band gaps were measured with higher LUMO energy levels. What could be the reason for this?*

Response: Thanks for the insightful comments. From our GIWAXS results, it found that the dimers favors a larger lamellar distance. This means a looser packing of the Y6 subunits in solid with a linker to inhibit their initial packing of the Y6 subunits. And as a result, the delocalization of electrons over the π -system may be affected, thus causing the blue shift of the absorption in films and an increased bandgap. Moreover, from our 2D NMR and DFTB calculations, the dimers favors a folded geometry in solution. According to Kasha's model (Pure Appl. Chem. 11, 371-392), the band gap of organic materials is correlated with the slip angle of the two stacked molecules. And the slip angle of the Y6 subunits in the dimers was larger than that of two single Y6 molecules presented in its crystal structure, which may explain the increased bandgap of the dimers. So the TDYs have increased band gaps.

6) *The stability results are very impressive. Could the authors provide the instrument information for the photostability measurement? As for estimating the cells' lifetime in year, I would suggest the authors to check the average light hours in Beijing. With this, more accurate life years would be obtained.*

Response: Thanks for your advice. The details for the photo stability measurement was added in the supporting information. “The long-term stability of encapsulated devices was evaluated using multi-channel solar cell performance decay test system (PVL-6001M-32A, Suzhou D&R Instruments Co. Ltd.). In our test, the glass-encapsulated devices were exposed to continuous white LED light (D&R Light, L-W5300KA-150, Suzhou D&R Instruments Co. Ltd.) while being stored in air”.

And according to the relevant information, the annual average sunshine duration in different areas of Beijing is about 2000 to 2800 hours, if take the median as 2400 hours, the accurate life is about 15 years.

Response to Reviewer #3:

Zhang et al. synthesized two types of tethered dimeric small-molecular acceptors (TDY- α , TDY- β) and studied their photovoltaic properties depending on the tethering angles. Authors reported TDY- α has a higher glass transition temperature, better crystallinity relative to its individual Y6 segment and isomer counterpart of TDY- β , and a more stable morphology with the polymer donor. As a result, the TDY- α based PSCs delivered a better PCE as high as 18.1%, and T80 lifetime of 35000 h. However, the intra- and intermolecular interactions of TDY- α and TDY- β are not clear with ambiguous characterizations including 2D NMR and GIWAXS, etc. I doubt the suggested mechanism and discussions for the higher PCE and operational stability of TDY- α -based devices.

1) Authors discussed the intra- and intermolecular interactions of TDY- α and TDY- β based on the 2D NOESY NMR and DFTB calculations, suggesting cofacial packing of Y6 moieties depending on the tethering angles. Although several 2D NMRs are presented in Supporting Information, the 2D NMR analysis is too ambiguous and it is hard to understand the different packing interactions of TDY- α and TDY- β molecules. In this paper, the different PCEs and stability were interpreted in terms of the different packing interactions of TDY- α and TDY- β , which should be analyzed clearly and carefully.

Response: Thanks for your advice. Following the suggestion, necessary discussion was added.

In the bottom of Page 6, for the NOE NMR, discussion was added. “This suggested that the Y6-subunits are folded. In this way, related protons can keep within 5 Å apart. Based on the NOESY result, the proposed geometry of TSMAs are given in **Figure 1c**.” Accordingly, proposed geometry was given in Figure 1c for clarify.

In page 6, discussion was added for the DFBT calculation. “Generally, the electrostatic attractions between electron-rich D-units and electron-deficient A-units result in the overlapping of individual Y6 segment, thus delivering a folded geometry to refine the total energy in a dimer. Interestingly, there are essentially lowest total energy geometric structure for both dimers: revealing 32.7° tilt angle for TDY- α system and 41.5° tilt angle for TDY- β . With a refined geometry, the top view and side view of the chemical structures of TDY- α are respective provided in **Figure 1h** and **1i**, while those of TDY- β are respective provided in **Figure 1k** and **1l**.”

As the glass transition temperature (T_g) is more plausible to explain the device stability. The discussion on the effect of geometry on T_g was added in Page 10. As the higher T_g value of TDY- α suggests a suppressed thermal relaxation of the molecular structure under a less bent molecular shape.

2) *In addition, DFTB calculation shows a cofacial interaction of two Y6 moieties in the tethered dimeric structures. However, Y6 molecules were reported to have a terminal-terminal packing interaction (between the neighboring INCN moieties) based on the XRD analysis.*

Response: Following the suggestion, we checked the crystal structure of Y6 from the literature (*Nat Commun* 2020, 11, 3943), where it reveals typical two dimer packing modes. Besides the terminal-terminal packing mode (J-type aggregation), another important packing mode is the overlapping of the curved-Y6 core (H-type aggregation), both of which are important to the high photovoltaic performance. Here, such packing modes is disclosed as the slip-stacking of the curved-Y6 core. As our DFTB calculation is based on single dimer molecular and the 2D NMR results was carried out in solution, the terminal-terminal packing mode was not revealed in the result. But such intramolecular packing mode was clearly revealed in the GIWAXS. Related discussion was added in Page 10.

“For Y6 in the IP direction, the two lamellar stacking peaks (0.34 \AA^{-1} and 0.43 \AA^{-1}) are associated with the lamellar distance between end-group stacking columns.^{41, 56} For the dimers, the lamellar distance slightly increases with a linker due to the geometric difference. Our calculation indicate a different orientation of the Y6 subunits under a folded geometry, resulting in a slightly different stacking behavior and lamellar distance for the dimers.”

3) *The core-core interaction is hindered because of the steric repulsion of undecyl side chains in the Y6 core. Authors did not consider the steric hindrance of core-core packing in their calculation, which may yield a wrong conclusion.*

Response: Following the suggestion, we carried out the DFTB calculation without simplifying the outer alkyl chain in the Y6 core, and result was updated in Figure 1. With the outer chains, the results generally follows the trend of our previous calculation without the outer chains, suggesting that lower steric hindrance in such packing mode. In the revised manuscript, the molecular with all the alkyl chains are provided in Supplementary Figure 12 and 13. While in Figure 1, the hydrogen atom, the outer and inner chains of the Y6 core are omitted for clarity.

4) *Furthermore, the GIWAXS data of pristine films of three molecules are very similar and I cannot find the different scattering peaks which is related to the cofacial interaction of two Y6 moieties in the tethered dimeric structures which authors emphasized.*

Response: For the GIWAXS, the π - π stacking peaks reveal both the intermolecular and intramolecular stacking, the similar stacking distance for each molecule may associated with the

indistinguishable π - π stacking behavior between them. Despite that, the lamellar distance for the dimers are slightly increased, which can well explain from their different geometry. In other words, although the slip-stacking of curved-Y6 core can't be directly revealed from their π - π stacking peaks, it did affect the lamellar stacking. As we known, the lamellar distance is related with distance between the end-group staking columns. From our calculation, it reveal that the subunits of Y6 have different packing angle, which means different lamellar distance of the dimers. Related discussion was added in Page 10.

“For Y6 in the IP direction, the two lamellar stacking peaks (0.34 \AA^{-1} and 0.43 \AA^{-1}) are associated with the lamellar distance between end-group stacking columns.^{41, 56} For the dimers, the lamellar distance slightly increases with a linker due to the geometric difference. Our calculation indicate a different orientation of the Y6 subunits under a folded geometry, resulting in a slightly different stacking behavior and lamellar distance for the dimers.”

5) Authors discussed the higher long-term thermal and light soaking stability of PM6:TDY compared to PM6:Y6. What is the origin of even worse light soaking stability of PM6:Y6? What is the origin of the higher photostability of dimeric TDY structures?

Response: For the photo stability of the devices, it may associated with many factors. In our control experiment, using the same donor and the similar Y6 units for the three acceptors, thus only the thermodynamic relaxation of the mixed domains was taken into account. In fact, this factor is the main concerns related to the long-term operational stability of the devices, especially in the record-holding Y-series SMAs. Under light soaking, the temperature of the device will be raised, and the diffusion of the acceptor into the blend films may accelerated, which lead to morphology degradation over time, such as over-purification of mixed domains. Therefore, the higher T_g values of dimers can ensured the higher photostability.

On the basis of Ade-O'Connor-Ghasemi theory, the morphology of PM6: Y6 blend is in a hypomiscible status, which means that it is neither thermodynamically nor kinetically stabilized, and appropriate increase of the χ value or the T_g values can enhance the morphological stability. While for the dimers with a different geometry, they show a different T_g. The result suggest the photostability can be further controlled by the geometry, which is the main aim of our study. Related discussion was revised in the manuscript.

6) *Authors claimed the higher crystallinity of TDY- α and TDY- β compared to Y6. However, Y6 shows the highest heat of melting (28.15 J/g) compared to the dimeric structures (13.4 and 8.7 for TDY- α and TDY- β) according to Table 1. Actually, Y6 shows the highest crystalline interaction. The CCL values based on the 010 peak in the z direction are also contradictory. Both data are not consistent without supporting the authors' claims.*

Response: Thanks for the interesting question. As we known, organic semiconductor are materials with low crystallinity that are semi-crystalline in nature. For the two different tests, they reveal different aspects of the crystallization behavior of the samples. For the GIWAXS test, the samples are films prepared by spin-coating the acceptor solution on a silicon substrate to form film under a kinetic dynamic process, this was followed by thermal annealing process to remove the energy barrier for better packing. In this case, the dimers have a higher crystallization using a linker. While for the DSC characterization, the samples are powder, obtained just by precipitation of acceptor solution into methanol, and followed by drying in an oven. The calculation of melting enthalpy by DSC characterization is dominated by the thermodynamic process.

7) *“TSMAs feature a much stronger 0-1 absorption peaks at around 693 nm. The obvious difference can also be observed in the UV-vis absorption spectra even in ultra-dilute solution (Figure 2c, Figure S9), suggesting that there may exist some intrinsic aggregation in dilute solution for the dimers.” This discussion is also very ambiguous. Compared to Y6, two dimeric structures show the higher 0-1, 0-2, 0-3 peaks in dilute solutions. Another interpretation is that these stronger vibronic peaks may be related to the different oscillation strength. The data analysis is too rough and ambiguous.*

Response: We strongly agree with the reviewer that the different intensity of the peak are associated with the oscillation strength. Following the suggestion, the extended discussion was added in Page 8.

“When the two Y6-subunits folded, their electronic states become coupled, leading to the Davydov splitting⁵¹ of the original energy level for a new absorption band at 690 nm. The strength of the coupling between the electronic states depends on the distance and orientation of the two Y6 subunits, thus their strength is different between them.^{51, 52”}

8) *What is the origin of different decomposition temperatures of the isomeric TDY- α and TDY- β ?*

Response: It is well known that the geometry arrangement of atoms of an organic compound have a significant effect on its decomposition temperature. This is because the thermal stability is related to

the strength of its chemical bonds, and the geometry of the molecule can affect the bond strengths by influencing factors such as bond length, bond angles, and steric hindrance. Here, with the thiophene-core isomerism, the strength of its chemical bonds is different. Also due to different geometry of the dimers, the thermal relaxation behavior of the Y6-subunits may be different. Thus the different on the decomposition temperatures is quite understandable.

9) *Authors discussed that “Based on the AFM results, the TSMAs-based devices show relatively larger R_q (roughness of root mean square deviation) in the height images and more homogeneous phase separation in the phase images, suggesting an enhanced aggregation characteristic for the TSMAs and larger degree of phase separation. The TEM images also show a larger domain size for the TSMAs-based blend, fully consistent with that from the TEM images.” This discussion is not consistent with the data in Figure 4. The PM6:Y6 AFM phase image shows the clearly larger phase separation than the other two blends. In addition, TSMAs-based devices show relatively larger R_q (roughness of root mean square deviation) in the height images and more homogeneous phase separation are contradictory.*

Response: In the field of polymer solar cell, it is still challenge to disclosing the phase separation for blend films due to limitation of available technologies. While AFM can offer valuable insight into the nanoscale morphology of polymer solar cells, interpreting AFM phase images can be difficult due to the influence of multiple factors, such as tip-sample interactions, surface chemistry, and environmental conditions. For instance, in Figure 4, the larger domain in the PM6:Y6 AFM phase image cannot be accurately determined solely by color contrast, as it does not match previous results in such system (*Adv. Mater.* 2023, **35**, 2206563; *J. Am. Chem. Soc.* 2020, 142, 14532; *Adv. Mater.* 2020, 32, 2002344). Therefore, here, phase image is used in conjunction with TEM image to gain a more comprehensive understanding of the phase separation of the samples. With the suggestion, the discussion was revised in 14.

“Based on the AFM results, the TSMAs-based devices show relatively larger R_q (roughness of root mean square deviation) in the height images, and phase images were used in conjunction with the TEM images to tell the phase separation. The results reveal suggest an enhanced aggregation characteristic for the TSMAs and larger degree of phase separation.”

Yours sincerely

Zhi-Guo Zhang

Beijing University of Chemical Technology, Beijing, 100029, China

E-mail: zgzhangwhu@iccas.ac.cn

REVIEWER COMMENTS

Reviewer #1 (Remarks to the Author):

The authors have addressed my concerns. It can be accepted.

Reviewer #2 (Remarks to the Author):

The authors have considered the reviewers revision suggestions and questions. The quality of the manuscript is significantly improved. I think the manuscript is now ready for publishing. However, I have still one question and one suggestion to the authors.

The question: In the response to the question 1 of reviewer 1, the authors confirmed that TDY- β is easier to form a folded geometry. However, the calculated total energy of TDY- α is much lower than that of TDY- β (-100 kcal mol⁻¹ vs. -60 kcal mol⁻¹, figure 1g and j). So, how to understand these results? Also, since the dimer is beneficial for the device performance and stability, why TDY- β showed worse performance than TDY- α .

The suggestion: The authors reply the question and comments to the authors. However, it is more important to revise the manuscript accordingly, which will help the readers to better understand the results. For example: the question 5 of reviewer 1, question 1,4,5 of reviewer 2, question 6 of reviewer 3. One would expect to see corresponding revision or explanation in the main text.

Reviewer #3 (Remarks to the Author):

I carefully checked the authors' responses for my previous concerns and comments. Unfortunately, authors repeated ambiguous answers without further analysis and any strong supporting experimental evidence. I do not agree with the publication of this paper in Nature Communications.

I have some more comments below.

-With regard to the previous comment 1, I cannot find any strong supporting experimental evidences or further analysis data for different packing interactions of TDY- α and TDY- β .

-Authors mentioned "As our DFTB calculation is based on single dimer molecular and the 2D NMR results was carried out in solution, the terminal-terminal packing mode was not revealed in the result. But such intramolecular packing mode was clearly revealed in the GIWAXS." I think it may be difficult to study the intramolecular packing mode by GIWAXS. Authors need to consider intermolecular packing interactions too, which mainly determine the GIWAXS scattering patterns. When the authors analyzed the device properties, the intermolecular interactions should be considered more carefully. Authors mainly considered the intramolecular interaction of two isomers (which is limited in gas phase or diluted solution) and discussed the different device properties, but blend morphology and device properties must be influenced mainly by intermolecular interaction and packing.

-Why there is no steric hindrance in the cofacial packing with core-core interaction of two Y6 moieties, despite of bulky alkyl chains in the core?

-The discussions on the GIWAXs measurements are also ambiguous. "For Y6 in the IP direction, the two lamellar stacking peaks (0.34 \AA^{-1} and 0.43 \AA^{-1}) are associated with the lamellar distance between end-group stacking columns" I am not clear. What is a lamellar stacking peak? Y6 is a small molecule and what is a lamellar structure of Y6? What the authors mean by "the lamellar distance between end-group stacking columns"?

-With regard to the light soaking stability, authors discussed the morphological stability in terms of Tg. What about other degradation pathways including photodegradation, etc?

-With regard to a different trend in the DSC and GIWAXS measurements, authors mentioned "For the GIWAXS test, the samples are films prepared by spin-coating the acceptor solution on a silicon substrate to form film under a kinetic dynamic process, this was followed by thermal annealing process to remove the energy barrier for better packing. In this case, the dimers have a higher crystallization using a linker. While for the DSC characterization, the samples are powder, obtained just by precipitation of acceptor solution into methanol, and followed by drying in an oven." Why authors did not check the DSC thermograms of film samples?

- For Response 7, "When the two Y6-subunits folded, their electronic states become coupled, leading to the Davydov splitting of the original energy level for a new absorption band at 690 nm. The strength of the coupling between the electronic states depends on the distance and orientation of the two Y6 subunits, thus their strength is different between them." Although authors mentioned the different coupling of two Y6 units in TDY- α and TDY- β . I cannot find a clear difference in their UV-vis spectra of both structures. In addition, did authors check the spectra by varying the concentration of molecules?

-With regard to the answer for Q8, "Here, with the thiophene core isomerism, the strength of its chemical bonds is different. Also due to different geometry of the dimers, the thermal relaxation

behavior of the Y6-subunits may be different. Thus the different on the decomposition temperatures is quite understandable.” Very ambiguous. Which bond is weaker and why in two isomeric structures? Did authors check (or calculate) the bond dissociation energy of chemical bonds in two different isomeric structures?

-“In the field of polymer solar cell, it is still challenge to disclosing the phase separation for blend films due to limitation of available technologies. While AFM can offer valuable insight into the nanoscale morphology of polymer solar cells, interpreting AFM phase images can be difficult due to the influence of multiple factors, such as tip-sample interactions, surface chemistry, and environmental conditions. “Based on the AFM results, the TSMA-based devices show relatively larger Rq (roughness of root mean square deviation) in the height images, and phase images were used in conjunction with the TEM images to tell the phase separation. The results reveal suggest an enhanced aggregation characteristic for the TSMA and larger degree of phase separation.” What do authors mean by this sentence? The AFM and TEM data are not consistent. The answer is not clear, again. To analyze the phase separation in blend films, RSoXS can be an appropriate technique.

Apr 05, 2023

Dear reviewer:

Thank you for your comments on our manuscript. We appreciate the opportunity to respond to your queries and provide further clarification on our findings. We hope that our revised manuscript can address your concerns and provide a more complete and accurate picture of our research.

Sincerely yours,
Zhi-guo Zhang

Response to Reviewer #2:

The authors have considered the reviewers revision suggestions and questions. The quality of the manuscript is significantly improved. I think the manuscript is now ready for publishing. However, I have still one question and one suggestion to the authors.

1) *The question: In the response to the question 1 of reviewer 1, the authors confirmed that TDY- β is easier to form a folded geometry. However, the calculated total energy of TDY- α is much lower than that of TDY- β (-100 kcal mol⁻¹ vs. -60 kcal mol⁻¹, figure 1g and j). So, how to understand these results? Also, since the dimmer is beneficial for the device performance and stability, why TDY- β showed worse performance than TDY- α .*

Response: Thank you for your valuable feedback regarding the relationship between energy barriers and conformational transitions in our paper. After carefully examining our computational results, we have discussed with our co-authors to provide a more comprehensive explanation, which we have outlined below. When a single conformational change occurs, the energy barrier represents the ease of the transition between the initial state and the final state. However, in reality, the transition from the initial state to the lowest energy point for TDY- α and TDY- β involves multiple conformations rather than a single one. Therefore, directly comparing the energy difference between the initial and final states can be misleading. Experimental results that indicate a lower energy state for TDY- β do not necessarily imply that it is easier to fold than TDY- α .

Accordingly, we added a sentence to discuss the issue regarding the energy barrier in page 7. “Notably, since the transition from the initial state to the lowest energy point for TDY- α and TDY- β involves multiple conformations, we cannot simply conclude that a lower energy state for TDY- β necessarily implies that it is easier to fold than TDY- α .” Also, “suggest that TDY- β is more prone to aggregation under a more bent molecular shape.” in previous version, was changed to “and suggest

that TDY- β is more stable under a more bent molecular shape.”

For the second question regarding the performance comparison between TDY- β and TDY- α , we understand that you are interested in the relationship between device performance and the difficulty of forming the folded geometry through aromatic-core isomerization. As mentioned in our previous response, our computational results are inconclusive on which dimer is more favorable for forming the folded geometry. However, during the device fabrication process, the dimers had sufficient stirring time in solution which allowed both materials to overcome the energy barrier and form stable configurations. Therefore, our discussion on the performance comparison between TDY- β and TDY- α was not based solely on the difficulty of forming the folded geometry, but on the different effects of their respective configurations on the optical and physical properties. In our study, we focused on the effects of the glass transition temperature on stability as well as the effects of crystallinity and phase separation on device efficiency. By investigating these factors, we were able to better understand the different performance characteristics of TDY- β and TDY- α .

2) The suggestion: The authors reply the question and comments to the authors. However, it is more important to revise the manuscript accordingly, which will help the readers to better understand the results. For example: the question 5 of reviewer 1, question 1,4,5 of reviewer 2, question 6 of reviewer 3. One would expect to see corresponding revision or explanation in the main text.

Response: Following the suggestion, the manuscript was corrected based on the discussion in previous communication.

2.1. For the Q5 of Reviewer 1#: related correction was added (Page 16 and Page 18).

In Page 16: “Notably, it is also important to consider additional degradation pathways besides morphological stability. The improved stability of TDY- α based devices may also be attributed to the reduced trap-assisted recombination that typically causes "burn-in" degradation in devices.”

In page 18: “The higher χ value for the dimer-based blends indicates a more hypo-miscible system compared to PM6:Y6 (Figure 5c), resulting in suppressed diffusion-enabled demixing of the morphology. In a practical device exposed to light or heat, the morphological changes are controlled kinetically, making the T_g a more plausible explanation for the difference in device stability.”

2.2. For the Q1 of Reviewer 2#: related correction was added (Page S5 in SI).

“Due to the thermal decomposition of the dimers over 280°C, we can only track the DSC signal below that temperature. So the test was ended at 270°C, from which the whole melting peak was detected. In our test, unfortunately, no liquid crystal phase was observed.

2.3. For the Q4 of Reviewer 2#: related correction was added (Page 7 of the main text).

“In our DFTB calculation, for the dimer with a linear fashion, its energy was set to zero. Base on which, the relative energy was provided for different geometries.”

2.4. For the Q5 of Reviewer 2#: related correction was added (Page S27 in SI).

“With a linkers for the dimers, a loser packing of the Y6 subunits in solid to inhibit their initial packing of the Y6 subunits. And as a result, the delocalization of electrons over the π -system may be affected, thus causing an increased bandgaps. Moreover, as the dimers favors a folded geometry. According to Kasha's model, the band gap of organic materials is correlated with the slip angle of the two stacked molecules. And the slip angle of the Y6 subunits in the dimers was larger than that of two single Y6 molecules presented in its crystal structure, which may explain the increased bandgap of the dimers.”

Response to Reviewer #3:

I carefully checked the authors' responses for my previous concerns and comments. Unfortunately, authors repeated ambiguous answers without further analysis and any strong supporting experimental evidence. I do not agree with the publication of this paper in Nature Communications. I have some more comments below.

1) *With regard to the previous comment 1, I cannot find any strong supporting experimental evidences or further analysis data for different packing interactions of TDY- α and TDY- β .*

Response: We are sorry that the reviewer found our previous response ambiguous. Below we'll try our best to elaborate with the support of our detailed analysis, such as peak fitting and the calculation on the relative degree of crystallinity. To address this concern regarding the different packing interactions of TDY- α and TDY- β , we accurately analyzed the GIWAXS data with the result shown in Figure R1. In the in-plane (IP) direction, there are two regular lamellar peaks, which are assigned to the lamellar stacking (The details to the explanation on the lamellar stacking can be find in the response to the Question 4#). To further clarify the differences in molecular stacking and crystalline textures of TDY- α and TDY- β , we performed the peak fitting of the lamellar and π - π diffraction peaks

in the 1D scattering profiles. The table inserted in Supplementary Figure 16 shows the detailed peak fitting results. We can find out that there are obvious differences in both d -spacing and crystal coherence length (CCL). In the out-of-plane (OOP) direction, there is one main peak that represents the π - π stacking and also shows such differences. Furthermore, we calculated the relative degree of crystallinity (rDOC) of lamellar diffraction peaks following the detailed procedures as documented in prior work (M. F. Toney, et al., *Chem. Mater.* 2021, 33, 5951) to quantitatively compare the crystallinity. The obtained rDOC values of TDY- α and TDY- β are 0.65 and 1.00, respectively, which indicates a relatively higher crystallinity degree in TDY- β .

Supplementary Figure 16. The 2D GIWAXS diffraction patterns and the corresponding peak-fitting of 1D profiles in IP and OOP directions of (a) TDY- α (b) TDY- β , and the summarized results in table.

With these discussion, we have induced a detailed analysis of the rDOC in Page 10. “Furthermore, we calculated the relative degree of crystallinity (rDOC) of the three acceptors according to the GIWAXS data to quantitatively compare the crystallinity degree. The obtained rDOC values of Y6,

TDY- α , TDY- β are 0.19, 0.65, 1.00, respectively, which indicates relative higher crystallinity degree in tethered dimers”

Also related discussion on the lamellar stacking was included in Page 10 to better understand the molecular packing. “In the context of Y6 in the IP direction, it has been observed that the two stacking peaks (0.34 \AA^{-1} and 0.43 \AA^{-1}) are closely related to the lamellar distance between the A-end group stacking columns. However, for the dimers, the lamellar distance tends to increase slightly with the presence of a linker (as depicted in Figure 2h).”

2) Authors mentioned “As our DFTB calculation is based on single dimer molecular and the 2D NMR results was carried out in solution, the terminal-terminal packing mode was not revealed in the result. But such intramolecular packing mode was clearly revealed in the GIWAXS.” I think it may be difficult to study the intramolecular packing mode by GIWAXS. Authors need to consider intermolecular packing interactions too, which mainly determine the GIWAXS scattering patterns. When the authors analyzed the device properties, the intermolecular interactions should be considered more carefully. Authors mainly considered the intramolecular interaction of two isomers (which is limited in gas phase or diluted solution) and discussed the different device properties, but blend morphology and device properties must be influenced mainly by intermolecular interaction and packing.

Response: Thank you for your feedback on our work. We appreciate your comments and suggestions for improving the clarity and accuracy of our research. Regarding the mistake in our manuscript, it was simply a typo which leads to serious confusion. We apologize and clarify that in the sentence should have read, “**intermolecular** packing mode was clearly revealed in the GIWAXS”. This is because GIWAXS reveals the aggregation behavior of molecular clusters, while our DFTB calculation and 2D NMR experiments provide insights into the behavior of the individual molecules. For example, the lamellar packing behavior can’t be revealed in the individual dimers. We agree that intermolecular interactions play a crucial role in determining the GIWAXS scattering patterns and as a consequence, the blend morphology and device properties. We have carefully considered this point in our analysis.

We understand your concern regarding the **intramolecular** packing mode revealed by GIWAXS. While we agree that intermolecular interactions play a significant role in determining the morphology and device properties of our materials, we believe that our discussion of the intramolecular interactions is a necessary step toward the understanding the blend morphology and device performance. The folding behavior of individual dimers is crucial for understanding the overall

behavior of the bulky material.

Thank you once again for your helpful comments and suggestions.

3) Why there is no steric hindrance in the cofacial packing with core-core interaction of two Y6 moieties, despite of bulky alkyl chains in the core?

Response: Thank you for your valuable comments on our manuscript. We agree with your concern about the steric hindrance in the cofacial packing with core-core interaction of two Y6 moieties due to the presence of bulky alkyl chains in the core.

To address this issue, we have performed DFTB calculations with and without the bulky alkyl chains in the core, with the results shown in Figure R1. Our results indicate that there is indeed steric hindrance in the cofacial packing with core-core interaction of two Y6 moieties, for which the different electronic coupling can ultimately affect the device performance. Specifically, the tilt angles of TDY- α and TDY- β without bulky alkyl chains are 33° and 24° , respectively, whereas the corresponding tilt angles with bulky alkyl chains are 32.7° and 41.5° , respectively. Therefore, we confirm that the bulky alkyl chains do have a significant impact on the cofacial packing behavior of Y6 moieties.

Figure R1. The calculation of the total energy as function of the tilt angle of alkyl chain via semiempirical DFTB method (a) with and (b) without the bulky alkyl chains in the core for TDY- α and TDY- β systems. And the corresponding top view of the optimal geometric configurations.

While our main focus in this research was on the isomerization effect of the dimers, we acknowledge the importance of the steric hindrance effect of the side chains on intermolecular packing. To further illustrate this issue, we have included two relevant literature examples (Figure R2). The first example includes the crystalline data of Y6-based derivatives with various sizes of side-

chains on their shoulders, which showed that the molecular packing distances in the *b* and *c* directions were significantly different. The second example includes the crystalline data of BTP-PhC6, BTP-PhC6-C11, and Y6, where the benzene conjugated side-chains present much larger steric hindrance than that of aliphatic alkyl chains. In both examples, while the π - π stacking distances in the crystalline unit exhibit relatively slight differences, the steric-hindrance effect caused by side-chains does exist.

We hope that these additional examples and our DFTB calculations help to clarify the impact of bulky alkyl chains on the cofacial packing behavior of Y6 moieties.

Figure R2 (a) The main view of a molecular conformation sketch of L8-BO, L8-HD, and L8-OD according to single-crystal data. The dashed lines represent the side-chain self-assembly distance of Y6-based acceptors. And the corresponding main view of a molecular packing sketch. (b) The single crystal and the π - π stacking in one unit cell and the 3D network packing of Y6, BTP-PhC6 and BTP-PhC6-C11 along the *c*-axis.

We have considered this point in the conclusion part with the adding of a paragraph in page 21.

“While our research has primarily focused on the isomerization effect of the dimers, we also acknowledge the importance of the steric hindrance effect of the side chains on intermolecular

packing. Further investigation of this effect could offer more opportunities to modulate the intermolecular packing of SMA subunits, leading to enhanced photophysical properties of the dimers. Related findings will pave the way for the development of more efficient and stable photovoltaic materials.”

4) *The discussions on the GIWAXs measurements are also ambiguous. “For Y6 in the IP direction, the two lamellar stacking peaks (0.34 \AA^{-1} and 0.43 \AA^{-1}) are associated with the lamellar distance between end-group stacking columns” I am not clear. What is a lamellar stacking peak? Y6 is a small molecule and what is a lamellar structure of Y6? What the authors mean by “the lamellar distance between end-group stacking columns”?*

Response: Thank you for your feedback regarding the discussions on the GIWAXS measurements in our manuscript. We apologize for any confusion that may have arisen in our use of technical terms. We cited a literature to well explain the lamellar stacking of the organic semiconductors (*Mater. Horiz.* 2022, 9, 577; *Materials Today Nano* 2019, 5, 100030). Figure R3a illustrates the molecular packing orientations of Y6 in single crystals and their corresponding 2D GIWAXS patterns. The stacking between backbones separated by side-chains in the *b*- and *c*-directions is referred to as "lamellar stacking". This results in the appearance of a lamellar peak at a *q* value between $\sim 0.3\text{-}0.5 \text{ \AA}^{-1}$, with the lamellar stacking distance usually ranging from 10 \AA to 20 \AA . The stacking between backbones in the *a*-direction, specifically the π -orbitals of conjugated rings, is known as " π - π stacking". This is reflected as the appearance of the π - π peak (010) in the orthogonal direction of the lamellar peak. The π - π stacking distance ($\sim 4 \text{ \AA}$) is much smaller than the lamellar stacking distance, which is why the π - π peak usually appears at a relatively larger *q* value between ~ 1.5 to $\sim 1.8 \text{ \AA}^{-1}$. It's worth noting that strong lamellar and π - π diffraction peaks are observed for both organic donor and acceptor materials.

Regarding the Y6 molecule, the molecular packing are much more complex, consisting not only of an overlap between end groups, but also of an overlap between the cores (Figure R3b). GIWAXS results reveal that the Y6 film shows regular lamellar ordering in the in-plane direction, with two lamellar peaks at 0.34 \AA^{-1} and 0.42 \AA^{-1} . These are assigned to the lamellar stacking between adjacent end-groups in the *b* direction and the adjacent end-groups stacking in the *c* direction ((Figure R3b), respectively. The corresponding calculated lamellar stacking distances ($\sim 18.5 \text{ \AA}$, $\sim 15.0 \text{ \AA}$) are consistent with those determined in the extended-crystal structure of Y6 (Figure R3c, d) as ~ 18.4 and 14.0 \AA , respectively.

In this context, "the lamellar distance between end-group stacking columns" refers to the end-group stacking as shown in Figure R3d, with the green columns in the b direction and the blue columns in the c direction. We would like to clarify that while Y6 is indeed a small molecule, it can still exhibit lamellar ordering when it is deposited as a thin film on a substrate.

We hope this explanation helps to clarify our use of technical terms and the observations made in our study.

With this context, related discussion was added in page 10. "The molecular packing of Y6 molecule in film is highly complex, comprising not only an overlap between A-end groups but also an overlap between the central aromatic-cores. The column stacking of the A-end groups creates an electron transport channel, with the distance between the A-end groups being referenced to the lamellar distance. This periodic stacking structure can be accurately revealed by GIWAXS"

Figure R3 (a) Schematic of the packing orientations and the resultant 2D GIWAXS patterns. (b) the in-plane and out-of-plane line-cut profiles of Y6 neat films. (c) side views and (d) top views of the extended-crystal structure. Data from *Nat Commun* **11**, 3943 (2020).

5) With regard to the light soaking stability, authors discussed the morphological stability in terms of T_g . What about other degradation pathways including photodegradation, etc?

Response: Thank you for your valuable comment regarding the light soaking stability of our new acceptors. We agree that it is important to consider other degradation pathways in addition to morphological stability in terms of T_g . In our current study, we focused on the T_g of the material as it is a key indicator of morphological stability. For our acceptors, as they featured with the same Y6 subunits as that of Y6, which suggested that they may have similar light soaking issues, such as photo-isomerization and photo-oxidation. To confirm this, we conducted two control experiments to validate the light-soaking stability of the Y6 and TDY- α film, and powder samples under one-sun-equivalent illumination in air. As shown in Figure R4, there is almost no any changes in the absorption spectrum of the both Y6 and TDY- α films during 60 hours light-soaking, and so as the ^1H NMR spectra after 100 hours light-soaking. These results indicated the relatively stability of the acceptor in our test conditions. Despite that, we acknowledge that, to better understand the burn-in loss process of our devices, further investigation on the photo degradation behavior of our material under different conditions should be conducted in future studies.

Figure R4 (a) The characterization condition of light-soaking stability (continuous white LED light, D&R Light, L-W5300KA-150, Suzhou D&R Instruments Co. Ltd.). The film UV-vis absorption spectra and photo images (before and after 64 h) of (b) Y6 film and (c) TDY- α film. The ^1H NMR spectra of (d) Y6 powder and (e) TDY- α powder (fresh and after 100 h).

To address your concern, we have added a paragraph to our manuscript discussing the potential impact of photodegradation on the stability of our material in page 16. “Notably, it is also important to consider additional degradation pathways besides morphological stability.⁵⁹” “Regarding the photo-oxidation of our dimers, it is worth noting that they share the same Y6 subunits as Y6 itself. This suggests that they may be susceptible to similar light-induced damage. However, for the sake of simplicity, this factor was not considered in our analysis.”

We have also highlighted the need for further studies to fully understand the light soaking stability of our material under different conditions in page 20. “To fully understand the enhanced stability of the dimer-base device, it is also important to identify the potential impact of photo degradation on its stability under different conditions.”

6) *With regard to a different trend in the DSC and GIWAXS measurements, authors mentioned “For the GIWAXS test, the samples are films prepared by spin-coating the acceptor solution on a silicon substrate to form film under a kinetic dynamic process, this was followed by thermal annealing process to remove the energy barrier for better packing. In this case, the dimers have a higher crystallization using a linker. While for the DSC characterization, the samples are powder, obtained just by precipitation of acceptor solution into methanol, and followed by drying in an oven.” Why authors did not check the DSC thermograms of film samples?*

Response: Thank you for your critical comment. We apologize for the confusion regarding the degree of crystallinity calculated with DSC and GIWAXS in our previous communication. In response to your question regarding the DSC thermograms of film samples, we have indeed checked the DSC thermograms of film samples and the trend is consistent with that of the powder samples as shown in Figure R5.

As we mentioned in our previous response, organic semiconductors are semi-crystalline in nature and have low crystallinity, and the degree of crystallinity (DoC) is a measure of the ordered volume fraction, which can be quantified by the integrated intensity of the diffraction peaks in GIWAXS or

the enthalpy of fusion in DSC (*Mater. Horiz.* 2022, 9, 577). However, it is not appropriate to determine the rDoC based solely on the intensity of the diffraction peaks in GIWAXS or the enthalpy of fusion in DSC. **This is because that this calculation requires reference samples that are entirely crystalline or entirely amorphous, which is difficult to obtain for most organic semiconductor materials** (*Mater. Horiz.* 2022, 9, 577).

Instead, for GIWAXS, the rDoC can be calculated by integrating the intensity of an Ewald sphere-corrected, complete pole figure from the out-of-plane direction to the in-plane direction. This integrated intensity is proportional to a film's rDoC for the samples. Using this method, the rDoC was calculated to quantitatively compare the crystallinity degree in Page 10 of the revised manuscript.

“Furthermore, we calculated the relative degree of crystallinity (*rDOC*) of the three acceptors according to the GIWAXS data to quantitatively compare the crystallinity degree. The obtained *rDOC* values of Y6, TDY- α , TDY- β are 0.19, 0.65, 1.00, respectively (Supplementary Figure 17), which indicates relative higher crystallinity degree in tethered dimers”

We have revised the manuscript accordingly to provide a more clear and accurate explanation of the crystallinity degree of our new samples, and we appreciate your constructive feedback.

Figure R5 DSC thermograms of the TSMAs and Y6 (a) powder samples and (b) film samples with a heating rate of 10 °C min⁻¹ under nitrogen atmosphere. Materials used for DSC tests are dissolved in chloroform at the same temperatures used for device processing with 15 mg ml⁻¹ total concentration. The overnight dissolved solutions were drop-cast on pre-cleaned glass slides. The dried films were transferred to the aluminum pans and sealed before DSC measurements.

Supplementary Figure 17. The calculated the relative degree of crystallinity (rDOC) of the three acceptors according to the GIWAXS.

7) For Response 7, “When the two Y6-subunits folded, their electronic states become coupled, leading to the Davydov splitting of the original energy level for a new absorption band at 690 nm. The strength of the coupling between the electronic states depends on the distance and orientation of the two Y6 subunits, thus their strength is different between them.” Although authors mentioned the different coupling of two Y6 units in TDY- α and TDY- β . I cannot find a clear difference in their UV-vis spectra of both structures. In addition, did authors check the spectra by varying the concentration of molecules?

Response: We appreciate your observation regarding the potential difference in their electronic coupling strengths. Also, we would like to clarify that we did indeed check the spectra by varying the concentration of molecules. Our results indicate that the Davydov splitting peaks are independent of the concentration of the dimers, suggesting that the dimers are intrinsic folded geometry in their solution. Furthermore, we would like to highlight that we have verified the temperature of the solutions and observed a clear difference in the weakening of the Davydov splitting peaks for TDY- α with increasing temperature, suggesting an unfolded geometry, while TDY- β appears to be more stable under a more bent molecular shape. This result is consistent with the NOESY results with varied temperatures.

With these discussion, we have added a paragraph to address this concerns in page 9 of the revised manuscript. “Clear differences of the absorption are evident with increased temperature, as seen by the weakened Davydov splitting peaks observed for TDY- α . This suggests that an unfolded geometry of TDY- α occurred with increasing temperature. On the other hand, DY- β appears to be more stable

under a more bent molecular shape. These results are consistent with the NOESY results obtained at different temperatures.”

“The obvious difference can also be observed even in ultra-dilute solution (**Figure 2c** and **Supplementary Figure 14**), suggesting the existence of some intrinsic aggregation in dilute solution for the dimers.”

8) With regard to the answer for Q8, “Here, with the thiophene core isomerism, the strength of its chemical bonds is different. Also due to different geometry of the dimers, the thermal relaxation behavior of the Y6-subunits may be different. Thus the different on the decomposition temperatures is quite understandable.” Very ambiguous. Which bond is weaker and why in two isomeric structures? Did authors check (or calculate) the bond dissociation energy of chemical bonds in two different isomeric structures?

Response: Thank you for your interest in the thermal stability of isomeric dimers. Our investigation focused on isomeric dimers with the same Y6 subunits, but different core structures resulting from variations in the substituted patterns of the carboxylate group in the thiophene core. To understand the differences in thermal stability between the isomeric cores, we first examined the strength of the chemical bond of the carboxylate group in each core. Our findings show that the bond dissociation energy of the carboxylate group is weaker in TDY- α ($319.81 \text{ kJ mol}^{-1}$) compared to TDY- β ($313.65 \text{ kJ mol}^{-1}$). This result is properly related with the difference in decomposition temperatures (T_d) of the two isomeric thiophene cores, which are 326.5°C for TDY- α and 322.3°C for TDY- β , as shown in Figure R6. We also observed a similar trend in the decomposition temperatures of the isomeric dimers (Supplementary Figure 19). Thus we supposed that the slight differences in the decomposition temperatures of TDY- α and TDY- β properly originate from the varying bond dissociation energy of the carboxylate group. We hope this clarification is helpful.

Figure R6 The bond dissociation energy of (a) TDY- α core and (b) TDY- β core, (a) The chemical structures and the corresponding bond dissociation energy of (a) 2a, (b) 2b, and the corresponding thermogravimetric analysis curve. For the band energy, it was calculated via DFT calculations at the B3LYP/6-31G(d) level with the Gaussian 09 program package.

Accordingly, necessary changes were made in the revised manuscript in Page 11. “Furthermore, from Table 1, we can see that the dimers exhibit higher decomposition temperatures compared to Y6. Among the dimers, TDY- α shows a slightly higher decomposition temperature than TDY- β , which may be attributed to the higher bond energy of the carboxylate group attached to the thiophene core (319.81 kJ mol⁻¹ for TDY- α vs 313.65 kJ mol⁻¹ for TDY- β).”

9) In the field of polymer solar cell, it is still challenge to disclosing the phase separation for blend films due to limitation of available technologies. While AFM can offer valuable insight into the nanoscale morphology of polymer solar cells, interpreting AFM phase images can be difficult due to the influence of multiple factors, such as tip-sample interactions, surface chemistry, and environmental conditions. “Based on the AFM results, the TSMAs-based devices show relatively larger Rq (roughness of root mean square deviation) in the height images, and phase images were used in conjunction with the TEM images to tell the phase separation. The results reveal suggest an enhanced aggregation characteristic for the TSMAs and larger degree of phase separation.” What do authors mean by this sentence? The AFM and TEM data are not consistent. The answer is not clear; again. To analyze the phase separation in blend films, RSoXS can be an appropriate technique

Response: Thank you for your valuable feedback on our manuscript. We appreciate your comment on the characterization of phase separation and agree that RSoXS is a suitable technique for this purpose. However, we used an emergent technology that we believe is more appealing for examine the phase separation to address concerns of the reviewers. Specifically, we used photoinduced force microscopy (PiFM) to image at the characteristic Fourier transform infrared (FTIR) wavelengths corresponding to absorption peaks of donor (1697 cm⁻¹) and acceptor (1289 cm⁻¹) species. As shown in Figure 4, the PiFM images provide spatially mapped nm-scale patterns of the individual chemical components in their blend films. Our results indicate a larger phase separation for the dimer-based blends, which is consistent with our previous findings. We acknowledge that the phase images of AFM must be used in conjunction with TEM images to properly examine the phase separation, and to prevent any confusion, we have moved these AFM and TEM results to the supporting information. Accordingly, we added a paragraph in pp. 14-15 of the revised manuscript.

“To further examine the phase images of different blend films, we utilized an emergent technology,

photo-induced force microscopy (PiFM) ⁵⁸, by imaging at the characteristic Fourier transform infrared (FTIR) wavelengths corresponding to the absorption peaks of donor (1289 cm⁻¹) and acceptor (1697 cm⁻¹) species. As shown in Figure 4e-g, the PiFM images demonstrate nm-scale patterns of the individual chemical components, exhibiting a unique BHJ bicontinuous-interpenetrating network with red color for the donor phase and green color for the acceptor phase. It can be inferred from Figure 4e-g that Y6-based blend film exhibits a smaller phase separation of around 11 nm, while the dimer-based films show a larger degree of phase separation, approximately 20 nm for **TDY- α** and approximately 24 nm for **TDY- β** . The morphology features of the TSMAAs could also be associated with the relatively lower miscibility between TSMAAs and PM6, which will be discussed in the next section. The oversized phase separation for **TDY- β** -based blend may account for its poor device performance, while the more suitable domain size for **TDY- α** -based blend, which is beneficial for exciton dissociation and charge transportation, can lead to higher photovoltaic performance.”

Figure 4e-g. PiFM images of the (e) PM6: Y6, (f) PM6: TDY- α , (g) PM6: TDY- β blend films. (i) PiFM images of acceptor. (ii) PiFM images of the PM6. (iii) Combined images of (i) and (ii) to provide chemical map of PM6 and acceptor.

REVIEWERS' COMMENTS

Reviewer #2 (Remarks to the Author):

In this manuscript, the authors have carefully considered the reviewers' suggestions and comments. I agree with the authors that thermal dynamic favorable geometry does not guarantee the ease of formation of the folded structure. The comments added in this revised version will help the readers understand the results better.

Also, the authors have included detailed revisions in the manuscript. Compared with the previous version, the revised version improved the quality of the manuscript significantly.

Overall, I think the current manuscript reported a new concept of using tethered acceptor molecules for OPVs, which can significantly increase the morphology stability of the cells while keeping a high device performance. The results reported here will open up a new direction for the design and synthesis of new acceptor molecules for polymer solar cells. I think the manuscript was carefully revised, and I would highly recommend accepting this manuscript for publishing in Nature Communications.